# The Graphon Limit Hypothesis: Understanding Neural Network Pruning via Infinite Width Analysis

**Hoang Pham[1], The-Anh Ta[2], Tom Jacobs[3], Rebekka Burkholz[3], Long Tran-Thanh[1]**
[1] University of Warwick, [2] CSIRO's Data61, [3] CISPA Helmholtz Center for Information Security
hoang.pham@warwick.ac.uk, theanh.ta@csiro.au, tom.jacobs@cispa.de,
burkholz@cispa.de, long.tran-thanh@warwick.ac.uk

## Abstract

Sparse neural networks promise efficiency, yet training them effectively remains a fundamental challenge. Despite advances in pruning methods that create sparse architectures, understanding why some sparse structures are better trainable than others with the same level of sparsity remains poorly understood. Aiming to develop a systematic approach to this fundamental problem, we propose a novel theoretical framework based on the theory of graph limits, particularly graphons, that characterizes sparse neural networks in the infinite-width regime. Our key insight is that connectivity patterns of sparse neural networks induced by pruning methods converge to specific graphons as networks' width tends to infinity, which encodes implicit structural biases of different pruning methods. We postulate the *Graphon Limit Hypothesis* and provide empirical evidence to support it. Leveraging this graphon representation, we derive a *Graphon Neural Tangent Kernel (Graphon NTK)* to study the training dynamics of sparse networks in the infinite width limit. Graphon NTK provides a general framework for the theoretical analysis of sparse networks. We empirically show that the spectral analysis of Graphon NTK correlates with observed training dynamics of sparse networks, explaining the varying convergence behaviours of different pruning methods. Our framework provides theoretical insights into the impact of connectivity patterns on the trainability of various sparse network architectures.

## 1 Introduction

Deep neural networks have achieved remarkable success in a wide range of machine learning tasks, including computer vision [33, 18], natural language processing [63, 17], and scientific modeling [38]. A key ingredient in this success is overparameterisation [4, 9, 37, 5], networks are often trained with many more parameters than are strictly necessary, which facilitates optimization and generalization. However, this overparameterisation introduces substantial computational and memory overheads [25, 61, 56, 69], hindering deployment in resource-constrained environments such as mobile/edge devices, embedded systems, or real-time applications.

To address these challenges, network pruning has emerged as a fundamental approach for compressing deep networks by removing redundant weights [41, 31, 23, 25]. By identifying and eliminating parameters that contribute little to the model's output, pruning can produce sparse subnetworks that retain high performance while offering significant efficiency gains. Empirically, the Lottery Ticket Hypothesis (LTH) [23] demonstrates that randomly initialised dense networks contain sparse subnetworks (winning tickets) that, when trained in isolation, can match or exceed the performance of the original network. This observation has sparked significant research efforts including iterative pruning methods for finding such winning tickets [24, 14, 13, 28], post-training pruning approaches [29], various Pruning at Initialisation (PaI) techniques [43, 62, 66, 56, 69], and dynamic sparse

training [51, 21, 44, 45]. Parallel to this empirical progress, theoretical work has sought to prove the existence of effective sparse neural networks [49, 55, 11, 10]. Other works [48, 22] attempt to understand the effectiveness of sparse neural networks by analyzing gradient flow during training. Through the lens of the Ramanujan graph, [64, 7, 35] indicate that maximizing the graph connectivity of sparse networks improves the performance of subnetworks. Recently, [74] have studied the training dynamics of randomly pruned networks via NTK in an infinitely wide neural network setting.

Despite these advances, a comprehensive theoretical framework for understanding neural network pruning and the training dynamics of general sparse neural networks remains elusive. A particularly challenging aspect is explaining why sparse networks are often difficult to train effectively [26]. In this paper, we aim to develop a systematic framework for analysing sparse networks obtained by PaI methods. While NTK and other large width limit approaches [37, 5, 42] provide valuable tools for analysing dense neural network training, they are not suitable for the sparse network settings due to a fundamental difficulty of defining the limit of sparse networks of different sizes.

To address this problem, we develop a novel approach by leveraging graph limit theory. In particular, a pruning method produces binary masks that determine the active connections between neurons in layers. These masks can naturally be interpreted as the adjacency matrices of graphs defined over neural network layers. As the width of the network increases, the size of these adjacency matrices grows, and their structure becomes more regular and well-defined. In this view, the sequence of pruning masks generated by increasingly wide networks forms a sequence of graphs.

We hypothesize that, in the infinite-width limit, this sequence of graphs converges to a graphon that serves as a limit object for dense or structured sparse graphs [47, 8]. Informally, any large graph can be viewed as a random sample from some underlying graphon $\mathcal{W}(u, v)$ encoding the probability of an edge between node indices $u$ and $v$ [57]. We propose the **Graphon Limit Hypothesis** for sparse neural networks obtained by a given pruning method in the large width limit and provide empirical evidence to support it. Informally, the hypothesis states that:

*Given a sparsity level, each network pruning method defines sequences of binary masks $(M_n^{(l)})$ that converges layer-wisely to graphons $\mathcal{W}^{(l)}$ in cut distance as the network width $n \to \infty$.*

This new perspective allows us to unify diverse pruning techniques under a common mathematical framework. For example, Random Pruning at initialisation produces subnets whose configuration is an Erdős–Rényi random graph, hence its graphon limit corresponds to a constant graphon, while other PaI methods produce graphons with different patterns. Crucially, these graphons can be used to define an infinite-width model, analogous to how fully-connected networks yield NTK in the limit [37, 5, 42]. We formalise this idea by introducing the **Graphon Neural Tangent Kernel** (Graphon NTK), a kernel that captures the infinite-width behaviour of a pruned network specified by a graphon.

Our framework offers a new approach for analysing sparse neural networks through their graphon limits, provides a theoretically grounded tool for understanding how pruning affects training dynamics, and offers a principled basis for comparing or designing pruning strategies via their associated NTK behaviour. This framework generalises previous work on randomly pruned networks [74] where Random Pruning corresponds to constant graphons. Our contributions are summarised as follows:

- We introduce the concept of graphon limits of pruning masks and propose the *Graphon Limit Hypothesis* that each pruning method corresponds to a distinct graphon in the infinite-width limit, supported by empirical evidence showing that pruning masks exhibit structural convergence and generate characteristic graphons.

- We formalise the Graphon NTK, a new kernel framework that combines pruning structure with infinite-width analysis, and show how it can be derived from the graphon representation of the mask, highlighting the key differences between NTK and Graphon NTK.

- We empirically establish a connection between the spectral properties of the Graphon NTK and training dynamics of sparse networks, providing theoretical insights into how connectivity patterns affect the trainability of sparse networks.

The remainder of this paper is organised as follows: Sections 2, 3 discuss related works and provide preliminaries on graph limits, NTK, and pruning methods, respectively. In Section 4, we formalise the Graphon Limit Hypothesis and provide empirical evidence for its validity. Section 5 derives the Graphon NTK and its special case of Random Pruning. Section 6 presents our experimental results. Section 7 concludes with a summary of our contributions and directions for future research.

## 2  Related work

**Neural network pruning.** Pruning has been extensively studied as a means of reducing the computational and memory demands of deep networks. Early works propose magnitude-based pruning, where weights with the smallest absolute values are removed after training [41, 32, 31]. The Lottery Ticket Hypothesis (LTH) [23] represents a paradigm shift by demonstrating that dense networks contain sparse subnetworks that can be trained in isolation to match or exceed the performance of the original network. Follow-up works expand this observation across architectures [14, 13] and training regimes [24, 28], while theoretical analyses [49, 55, 53, 10, 11] seek to establish formal conditions under which such "winning tickets" exist. However, finding these winning tickets typically requires computationally intensive iterative train-prune-retrain cycles. To address this computational burden, Pruning at Initialisation (PaI) methods have emerged, which identify sparse subnetworks before training begins. Subnetworks can be found based on the magnitude, gradient, or hessian information [43, 65, 16, 3] or NTK-based scores [46, 54, 30, 67], while other methods [62, 56, 69] try to identify these tickets based on the subnetworks' configuration only. Despite these advances, analyses of why different pruning methods yield varying performance at the same sparsity level remain largely empirical. Works such as [48, 22, 36] have analysed gradient flow in sparse networks, while [74] examines the NTK limit behaviour of randomly pruned networks. Our work advances these efforts by providing a theoretical framework for understanding the limiting behaviour of pruned networks as their width increases.

**Graphon and graph limit.** Graphons are limit objects of graph sequences and have become a central tool in the study of large networks [47, 8]. A graphon is a symmetric measurable function $\mathcal{W} : [0,1]^2 \to [0,1]$ that can be viewed as the continuum analogue of an adjacency matrix. Graphons have been widely used in, e.g., modelling social networks [60, 58], and community detection [39, 1]. In machine learning, graphons have found use in graph neural network (GNN) analysis. Graphon neural networks extend traditional GNNs by considering their behaviour in the limit of large graphs [57, 52, 50, 34]. In particular, graphon neural networks operate on functions over $[0,1]$ rather than on finite-dimensional vectors, which makes it possible to model infinitely large graphs or to generalise across graphs of different sizes. Another line of work focuses on learning a graphon from a collection of observed graphs [2, 12, 70, 68]. Specifically, to estimate graphons, SAS [12] reorders adjacency matrices by degree before applying smoothing, SBA [2] fits a stochastic block model to the graph data, GWB [70] relaxes the cut distance by Gromov-Wasserstein distance and minimizes this distance between observed graphs, and IGNR [68] directly uses neural networks. However, to the best of our knowledge, no prior work has connected graphons to the behaviour of pruning methods in neural networks. Prior studies on pruning GNNs [15, 59, 75] focus on removing edges in input data graphs to enhance computational efficiency or task performance. In contrast, we study pruning within the model's own weight graph, which form an implicit connectivity graph within the neural network architecture itself. Our work introduces a novel interpretation: pruning masks of increasingly wide neural networks define a sequence of binary graphs that converge to graphons, and these graphons encode structural priors in the infinite-width limit.

**Neural tangent kernel and infinite-width limit.** The neural tangent kernel (NTK) [37, 5, 42, 19] characterises the training dynamics of infinitely wide neural networks under gradient descent. In this regime, training a network becomes equivalent to solving a kernel regression problem using the NTK. The NTK has since been extended and studied across a wide variety of architectures, including CNNs, RNNs, transformers, GNNs [5, 71, 72, 40, 20]. Among these, Graph Neural Tangent Kernels (GNTKs) [20] allow study on the limiting behaviour of infinitely wide (the number of features) GNNs trained via gradient descent. Later, [40] demonstrates that as the size of the graph increases, the GNTK converges to a graphon neural tangent kernel [1]. While most NTK analyses assume dense architectures, recent studies have begun to explore sparse NTK models. [74] shows that Random Pruning preserves the NTK in the infinite-width limit up to a scaling factor, with convergence improving at larger widths. In particular, they apply pre-defined random masks on layers' weights to derive the NTK. Separately, [73] analyses sparsity induced by large bias initialisation, demonstrating that it leads to structured sparse activations and improved NTK conditioning, yielding faster convergence and tighter generalization bounds. Our work extends this line of research by developing a framework for analysing arbitrary pruning methods through their limiting graphons. This enables a more nuanced understanding of how different pruning strategies affect network trainability in the large width limit.

---

[1]This kernel, also named Graphon NTK in [40], is for graph data and is different from our kernel.

## 3 Preliminaries

### 3.1 Neural tangent kernel

The neural tangent kernel (NTK) [37, 19, 5, 42] provides a theoretical framework for understanding the training dynamics of overparameterized neural networks. For a network $f(x; \theta)$ mapping input $x \in \mathbb{R}^d$ to output in $\mathbb{R}$, the NTK is defined as: $\Theta(x, x') = \nabla_\theta f(x, \theta)^\top \nabla_\theta f(x', \theta)$.

In the limit where the width of each hidden layer tends to infinity and the parameters are initialised randomly (e.g., with Gaussian scaling), the NTK converges to a deterministic kernel $\Theta_0(x, x')$, independent of training: for any given $t \geq 0$, $\Theta_t(x, x') \to \Theta_0(x, x')$, as $n \to \infty$. More recent works have shown that even networks with finite width, and any depth follows the NTK behaviour [5, 42]. Under gradient flow on the mean squared error loss, the predictions evolve according to a linear differential equation, which admits the solution: $f(X; \theta_t) = f(X; \theta_0)e^{-\eta \Theta_0 t} + \left(I - e^{-\eta \Theta_0 t}\right) \hat{y}$, where $\hat{y}$ is the training label. This formula shows that the network predictions interpolate exponentially toward the training labels $\hat{y}$, with convergence behaviour governed by the spectral properties of $\Theta_0$. In particular, convergence is faster along directions corresponding to larger eigenvalues of the NTK.

Different architectures yield different limiting NTKs. For example, fully connected ReLU networks admit closed-form recursive formulas for the entries of $\Theta_0$ [37], while convolutional and residual architectures induce structured NTKs that encode translation invariance and hierarchical composition [5]. Thus, the NTK framework bridges deep learning and kernel methods, offering theoretical insights into optimisation and generalisation.

### 3.2 Graphon and graph limit theory

Graph limit theory provides an analytic framework for analysing large graphs. For a sequence of graphs with an increasing number of nodes $(G_n)$, one can often associate a limit object known as a graphon which is a symmetric, measurable function $\mathcal{W} : [0, 1]^2 \to [0, 1]$ that encodes the connectivity pattern between an infinite set of nodes [47, 8]. The function value $\mathcal{W}(x, y)$ represents the probability of an edge between nodes indexed by $x$ and $y$ in the limit. Graphon theory is particularly useful for capturing the limiting behaviour of a graph sequence $(G_n)$ as the number of nodes $n \to \infty$.

Each finite graph $G_n$ with $n$ vertices can be represented (up to relabelling) by a step function $\mathcal{W}_{G_n}$ on $[0, 1]^2$, where the unit interval is partitioned into $n$ equal parts and edge presence is encoded by: $\mathcal{W}_{G_n}(x, y) = A_{ij}, \forall (x, y) \in \left[\frac{i-1}{n}, \frac{i}{n}\right) \times \left[\frac{j-1}{n}, \frac{j}{n}\right)$, with $A$ being the adjacency matrix of $G_n$. A sequence $(G_n)$ is said to converge to a graphon $\mathcal{W}$ if the sequence $(\mathcal{W}_{G_n})$ converges to $\mathcal{W}$ in the labelled cut distance:

$$d_\square(\mathcal{U}, \mathcal{W}) = \sup_{S, T \subset [0,1]} \left| \int_{S \times T} (\mathcal{U}(x, y) - \mathcal{W}(x, y)) dx dy \right|,$$

where $\mathcal{U}$ and $\mathcal{W}$ are two graphons, and the supremum is taken over all measurable subsets $S, T \subseteq [0, 1]$. To account for vertex relabelling, we define the cut distance between two graphons up to measure-preserving transformations. Let $\Phi$ be the set of all measure-preserving bijections $\phi : [0, 1] \to [0, 1]$. The (label-invariant) cut distance between graphons $\mathcal{U}$ and $\mathcal{W}$ is modified to be: $\delta_\square(\mathcal{U}, \mathcal{W}) = \inf_{\phi \in \Phi} d_\square(\mathcal{U}, \mathcal{W}^\phi)$ where $\mathcal{W}^\phi(x, y) = \mathcal{W}(\phi(x), \phi(y))$. A sequence of graphs $(G_n)$ with $|V(G_n)| \to \infty$ is said to converge to a graphon $\mathcal{W}$ if $\lim_{n \to \infty} \delta_\square(\mathcal{W}_{G_n}, \mathcal{W}) = 0$.

The cut distance quantifies the similarity between two graph adjacency matrices or functions by considering all possible labellings of their nodes [47]. Intuitively, two graphs are close in the cut distance if their global connectivity structures are similar. An equivalent characterization of the convergence of a graph sequence $(G_n)$ to a graphon is based on homomorphism counting [8, 57]. Particularly, for every finite subgraph $F$, the homomorphism densities $t(F, G_n)$ converge to $t(F, \mathcal{W})$. The homomorphism density $t(F, G)$ represents the probability that a random map from the vertices of $F$ to those of $G$ preserves all edges.

## 4 Graph limit and sparse neural networks

Neural network pruning produces binary masks over the network's layers to specify which weights are removed from each layer. These masks naturally define bipartite graphs between adjacent layers, with the mask matrix $M^{(l)} \in \{0, 1\}^{n_{l-1} \times n_l}$ acting as the biadjacency matrix. We hypothesize that

pruning-induced structures can be modelled in the infinite-width limit using graphons, enabling a unified geometric and functional interpretation of sparse neural networks.

## 4.1 Pruning masks as graphs

In a fully connected network, each neuron in layer $l-1$ is connected to every neuron in layer $l$, resulting in a dense weight matrix $W^{(l)} \in \mathbb{R}^{n_{l-1} \times n_l}$. A pruning method replaces this weight matrix with a masked version $\widetilde{W}^{(l)} = W^{(l)} \odot M^{(l)}$ where $M^{(l)} \in \{0,1\}^{n_{l-1} \times n_l}$ is the binary mask and $\odot$ denotes elementwise multiplication. The binary mask $M^{(l)}$ defines a bipartite graph, and in the infinite-width limit, we model it with a bipartite graphon $\mathcal{W}^{(l)} : [0,1]^2 \to [0,1]$. From this point onward, we use $\mathcal{W}$ to denote a bipartite graphon, unless explicitly mentioned otherwise.

Despite being called "sparse", most pruning methods retain a constant fraction of weights (e.g., 10%), resulting in $\Theta(n^2)$ connections for width $n$ (although with very small constants). From a graph-theoretic perspective, these remain dense graphs, where graphon theory applies. Thus, to ensure well-defined graphon limits, we assume: (i) the pruning masks retain a constant fraction $p > 0$ of weights (normally satisfied in pruning context), and (ii) the width of hidden layers remains comparable across layers as $n \to \infty$. Under these assumptions, given a pruning method, for a sequence of pruning masks $(M_n^{(l)})$ with increasing width, we can define their convergence to a bipartite graphon $\mathcal{W}^{(l)}$ analogously: $\lim_{n \to \infty} \delta_\square(\mathcal{W}_{M_n^{(l)}}, \mathcal{W}^{(l)}) = 0$ where $\mathcal{W}_{M_n^{(l)}}$ is the bipartite graphon representation of the mask $M_n^{(l)}$ at layer $l$ (see Section 3.2 for definition of graphon representation of graphs). As the network width $n \to \infty$, each mask matrix becomes a larger and more structured graph. Different pruning methods induce different sequences $(M_n^{(l)})$ of such biadjacency matrices.

## 4.2 Graphon Limit Hypothesis for neural network pruning

**Graphon Limit Hypothesis.** *Given a sequence of neural networks $(N_n)$ from a fixed architecture class $\mathcal{A}$ with widths tending to infinity, the application of a pruning method $\mathcal{P}$ at fixed sparsity level $p > 0$ produces sequences of binary masks $(M_n^{(l)})$ that converge layer-wisely to deterministic graphons $\mathcal{W}^{(l)}$ in the cut distance. These limiting graphons depend only on $\mathcal{A}, p, \mathcal{P}$ and characterize the connectivity structure of the pruned networks in the infinite-width limit.*

The graphon limit has the following geometry interpretation. A fundamental result from graph limit theory states that a sequence of graphs $(G_n)$ converges to a graphon $\mathcal{W}$ if and only if the density of any fixed subgraph $F$ in $(G_n)$ converges to the density of $F$ in $\mathcal{W}$. In the context of neural networks, this means the graphon limit asymptotically captures geometric patterns including path densities (effective paths), and other structural motifs that influence the network's computational properties.

This hypothesis has several important implications: (i) graphons $\mathcal{W}^{(l)}$ serve as structural priors for the pruned network in the infinite-width setting; (ii) each pruning method corresponds to a distinct region in the space of graphons, determined by the method's design; and (iii) structural differences between pruning methods can be captured and compared through their limiting graphons.

## 4.3 Experiments on graph limit of pruning at initialisation methods

We empirically validate the Graphon Limit Hypothesis by examining whether pruning methods converge to distinct, characteristic graphons as network width increases. We analyse four pruning at initialisation methods: Random, SNIP [43], GraSP [65], and Synflow [62] across varying network widths $n \in \{100, 500, 1000, 2000\}$, number of layers $\{4, 5\}$, and sparsity levels $\{70\%, 80\%, 90\%\}$. We conduct 100 independent trials per configuration and collect layer masks except for masks from input and output layers. To visualise the emergent graphons, we employ the SAS method [12] that: (1) Sorts nodes based on degree centrality (out-degree for layer $l$, in-degree for layer $l+1$); (2) Partitions the sorted bipartite graph into a grid of intervals; (3) Computes the average edge density within each interval. This degree-based sorting serves as an approximate measure-preserving transformation, revealing underlying structural patterns while maintaining invariance to node permutations. We refer to Appendix C for more details.

Figure 1 displays the estimated graphons with increasing network widths. Each pruning method converges to a distinct pattern. In particular, Random Pruning converges to a constant graphon

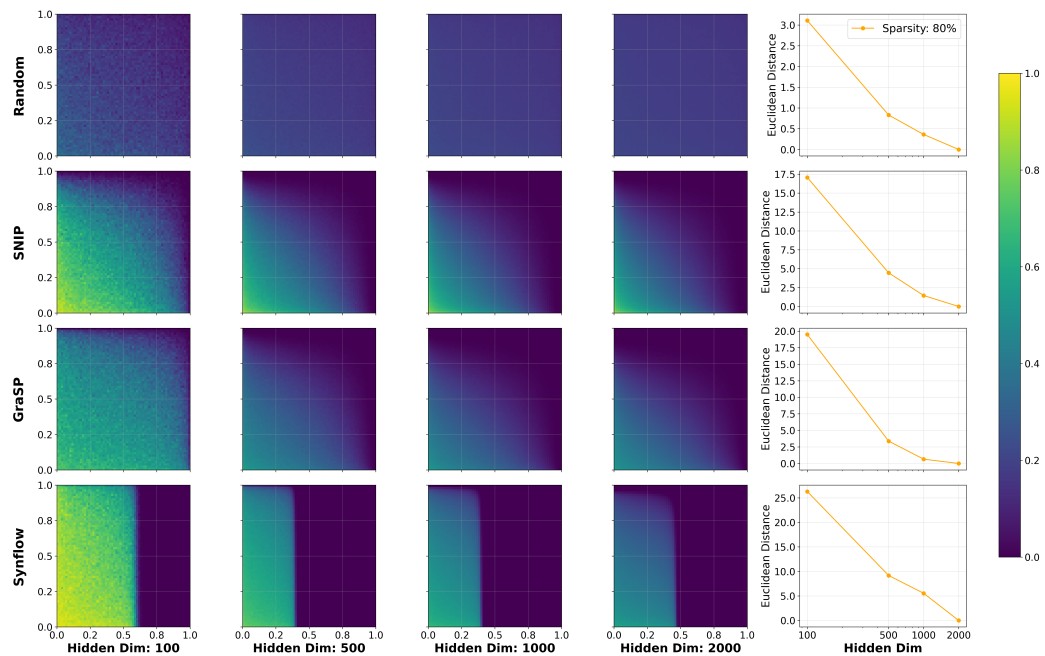

Figure 1: Graph limit of subnetworks' mask produced by PaI methods at 80% sparsity and the corresponding convergence of graphons via Euclidean distances.

(Erdős-Rényi random graph), with uniform connection probability across all node positions. SNIP and GraSP exhibit structured, non-uniform graphons with density gradients, preferentially connecting high-centrality nodes. Synflow converges to a block-like graphon with sharp transitions, strongly prioritising connections among high-centrality neurons. This method encourages sparse networks with a high number of paths, which is also indicated in [56, 54].

To quantify convergence, we also show the Euclidean distance between density matrices at width $n$ and reference matrices at $n = 2000$ in Figure 1-right. Since all histograms are aligned via degree-based sorting, we use the Euclidean distance between the density matrices as a proxy for the cut distance. All methods demonstrate monotonic convergence, with distances decreasing as width increases, confirming that the limiting graphon structure is an intrinsic characteristic of each pruning algorithm.

These results provide compelling evidence that each pruning method induces a unique (up to relabelling), stable connectivity pattern in the large-width limit, validating our graphon hypothesis and establishing a foundation for analysing pruning methods through graph limit theory.

## 5   Neural tangent kernel with graphon structure

In this section, we present the derivation of NTK for neural networks with graphon structure. This novel formulation extends the standard NTK theory to accommodate networks where connectivity patterns are modulated by graphon functions, providing insights into how sparse network architecture influences learning dynamics.

### 5.1   Network structure and setup

In standard neural networks, weights between layers are typically sampled from identical independent distributions. However, sparse networks exhibit connectivity patterns, where connection strength depends on neuron positions. Graphon provides a mathematical tool to model such connectivity patterns. Intuitively, we can view each neuron as having a position in $[0, 1]$, and the graphon value $\mathcal{W}^{(l)}(u, v)$ represents the expected connection strength/probability between neurons at positions $u$ and $v$. Each layer's structure is thus defined by its corresponding graphon $\mathcal{W}^{(l)}$, providing a continuous representation of the network's connectivity.

Graphon-structured networks naturally connect to neural network pruning, where certain connections are eliminated. In pruning, a binary mask $M^{(l)}$ is applied to weights: $\widetilde{W}^{(l)} = W^{(l)} \odot M^{(l)}$. For weights initialised as $W_{ij}^{(l)} \sim \mathcal{N}(0, \sigma_w^2)$, pruning effectively modifies their variance: $\text{Var}(\widetilde{W}_{ij}^{(l)}) = \text{Var}(W_{ij}^{(l)} M_{ij}^{(l)}) = \text{Var}(W_{ij}^{(l)}) M_{ij}^{(l)} = \sigma_w^2 M_{ij}^{(l)}$.

Based on the Graphon Limit Hypothesis in Section 4, as network width increases, pruning masks converge to graphons layer-wisely $\mathcal{W}^{(l)}(u_l, u_{l-1})$, representing the probability density of connections between positions. In a finite network, the weights are sampled as: $W_{ij}^{(l)} \sim \mathcal{N}\left(0, \mathcal{W}^{(l)}\left(\frac{i}{n_l}, \frac{j}{n_{l-1}}\right)\right)$, where $n_l$ represents the width of layer $l$, and we consider $\sigma_w^2 = 1$ for simplicity [5]. Similar to the standard NTK setting [37, 5], we remove the bias terms and consider a single output network. Unlike standard networks where weights are i.i.d., our graphon-modulated weights have position-dependent variances while maintaining independence. As we take layer widths to infinity $n_l \to \infty$, discrete neuron indices approach to continuous positions in $[0, 1]$: $i/n_l \to u_l \in [0, 1]$ (position in layer $l$) and $j/n_{l-1} \to u_{l-1} \in [0, 1]$ (position in layer $l-1$). Summations over neurons become integrals over positions, and the scaling term $1/n_{l-1}$ will be absorbed into the integral. The forward pass of an $L$ hidden layers network below illustrates this approach from discrete to continuous:

**Discrete network**

$$z_i^{(l)}(x) = \frac{1}{\sqrt{n_{l-1}}} \sum_{j=1}^{n_{l-1}} W_{ij}^{(l)} h_j^{(l-1)}(x),$$

$$h_i^{(l)}(x) = \sigma(z_i^{(l)}(x)),$$

$$f(x) = \frac{1}{\sqrt{n_L}} \sum_{j=1}^{n_L} W_{ij}^{(L+1)} h_j^{(L)}(x),$$

**Continuous network**

$$z^{(l)}(u_l, x) = \int_0^1 W^{(l)}(u_l, u_{l-1}) h^{(l-1)}(u_{l-1}, x) \, du_{l-1},$$

$$h^{(l)}(u_l, x) = \sigma(z^{(l)}(u_l, x)),$$

$$f(x) = \int_0^1 W^{(L+1)}(u_{L+1}, u_L) h^{(L)}(u_L, x) \, du_L,$$

where $W^{(l)}(u_l, u_{l-1}) \sim \mathcal{N}(0, \mathcal{W}^{(l)}(u_l, u_{l-1}))$ is a random field modulated by the graphon. In addition, since the statistical properties of the network now is affected by the graphon structure, weights are independent but *not identically* distributed random variables. We need additional assumptions to ensure the Law of Large Numbers (LLN) and Central Limit Theorem (CLT) still apply. For the LLN to hold in this non-iid setting, we assume: (i) the graphon values $\mathcal{W}^{(l)}(u_l, u_{l-1})$ are bounded, and (ii) the average connectivity $\int_0^1 \mathcal{W}^{(l)}(u_l, u_{l-1}) du_{l-1}$ are well-behaved for all positions $u_l$. Similarly, for the CLT to apply to pre-activations, we assume the Lindeberg-Feller [6] condition:

$$\lim_{n_{l-1} \to \infty} \frac{1}{n_{l-1}} \sum_{j=1}^{n_{l-1}} \mathbb{E}\left[\left(W_{ij}^{(l)} h_j^{(l-1)}(x)\right)^2 \cdot \mathbf{1}_{\{|W_{ij}^{(l)} h_j^{(l-1)}(x)| > \epsilon \sqrt{n_{l-1}}\}}\right] = 0,$$

for all $\epsilon > 0$. This condition ensures that no single weight-activation product dominates the sum, allowing the pre-activations to still converge to Gaussian processes, albeit with position-dependent covariance structures shaped by the graphon.

## 5.2 Graphon neural tangent kernel

We extend the NTK theory to networks with graphon-structured connectivity. We begin by characterising the limiting behaviour of pre-activations and activations in the infinite-width limit.

**Proposition 1.** *For a neural network with layers structured by graphons $\mathcal{W}^{(l)} : [0, 1]^2 \to [0, 1]$, Lipschitz nonlinearity $\sigma$, and in the limit as $n_1, ..., n_L \to \infty$, the pre-activations $z^{(l)}(u_l, x)$ at every hidden layer converge to centred Gaussian processes with covariance $\tilde{\Sigma}^{(l)}$, where $\tilde{\Sigma}^{(l)}$ is defined recursively by:*

$$\tilde{\Sigma}^{(1)}(u_1, u_1', x, x') = \delta(u_1 - u_1') \frac{1}{d} \sum_j^d \mathcal{W}^{(1)}(u_1, \frac{j}{d})(x \cdot x')_j, \tag{1}$$

$$\tilde{\Sigma}^{(l)}(u_l, u_l', x, x') = \delta(u_l - u_l') \int_0^1 \mathcal{W}^{(l)}(u_l, u_{l-1}) \Sigma^{(l-1)}(u_{l-1}, u_{l-1}, x, x') du_{l-1}, \tag{2}$$

*where $(x \cdot x')_j$ represents the input correlation at position $j$, the activation covariance $\Sigma^{(l)}$ is:*

$$\Sigma^{(l)}(u_l, u_l', x, x') = \delta(u_l - u_l')\mathbb{E}_{(z,z')\sim\mathcal{N}(0,\Lambda^{(l)}(u_l))}[\sigma(z)\sigma(z')], \qquad (3)$$

*$\delta(u_l - u_l')$ is the Dirac delta function, and $\Lambda^{(l)}(u_l)$ is the position-dependent covariance matrix:*

$$\Lambda^{(l)}(u_l) = \begin{bmatrix} \tilde{\Sigma}^{(l)}(u_l, u_l, x, x) & \tilde{\Sigma}^{(l)}(u_l, u_l, x, x') \\ \tilde{\Sigma}^{(l)}(u_l, u_l, x', x) & \tilde{\Sigma}^{(l)}(u_l, u_l, x', x') \end{bmatrix}.$$

**Remark 1.** *The key difference between the standard NTK and Graphon NTK lies in the pre-activation covariance formulation. In standard NTK, the pre-activation covariance $\tilde{\Sigma}^{(l)}$ equals the previous layer's activation covariance $\Sigma^{(l-1)}$. In contrast, the Graphon NTK modulates this with the graphon function $\mathcal{W}^{(l)}$, creating position-dependent covariance structures. This causes signals to propagate non-uniformly through the network, with connectivity strength determined by the graphon values.*

The NTK characterises how network outputs change with respect to parameters during training. Our key result shows that the Graphon NTK also converges to a deterministic limit in the infinite-width regime, but with a structure determined by the graphon connectivity patterns.

**Theorem 1** (Graphon NTK). *For a neural network with layers structured by graphons $\mathcal{W}^{(l)} : [0,1]^2 \rightarrow [0,1]$, Lipschitz nonlinearity $\sigma$, in the limit as $n_1, ..., n_L \rightarrow \infty$, the Graphon Neural Tangent Kernel (Graphon NTK) $\Theta(x, x')$ converges to a deterministic kernel:*

$$\Theta(x, x') = \sum_{l=1}^{L} \int_0^1 \left( \dot{\Sigma}^{(l)}(u_l, u_l, x, x') \int_{[0,1]^{L-l+1}} \prod_{m=l+1}^{L+1} \mathcal{W}^{(m)}(u_m, u_{m-1})\dot{\Sigma}^{(m)}(u_m, u_m, x, x') \, d\mathbf{u}_{l+1} \right)$$

$$\cdot \left( \int_0^1 \Sigma^{(l-1)}(u_{l-1}, u_{l-1}, x, x')du_{l-1} \right) du_l, \qquad (4)$$

*where $\dot{\Sigma}^{(l)}(u_l, u_l, x, x') = \mathbb{E}[\sigma'(z^{(l)}(u_l, x))\sigma'(z^{(l)}(u_l, x'))]$ represents the expected correlation between activation derivatives, and $d\mathbf{u}_{l+1} = du_{L+1}du_L \ldots du_{l+1}$.*

**Remark 2.** *The Graphon NTK explicitly shows how graphon functions $\mathcal{W}^{(l)}$ shape the kernel through position-dependent connectivity, in contrast to the standard NTK. This structure gives insights on the heterogeneous learning dynamics of sparse model training, and allows modelling diverse connectivity patterns.*

The Graphon NTK provides a powerful framework for analysing how the pattern of connectivity between neurons in sparse neural networks affects learning dynamics in the infinite-width limit. By modulating the weight variances according to the graphon, we can model a wide range of structured connectivity patterns, including those that arise from various network pruning strategies. We refer to Appendix A for detailed proofs.

### 5.3 Graphon neural tangent kernel of Random Pruning

Our Graphon NTK framework encompasses a broad class of structured sparsity patterns with Random Pruning being a special case. Specifically, when the underlying graphon is constant, i.e., $\mathcal{W}(u, v) = c$ where $c \in (0, 1]$, the resulting kernel simplifies to a scaled version of the standard NTK: $\Theta(x, x') = c^L \Theta_{\text{std}}(x, x')$, where $\Theta_{\text{std}}$ denotes the standard NTK of a fully-connected network with $L$ hidden layers. This special case was previously studied in [74] by applying random masks to weight matrices. Our general framework of Graphon NTK not only recovers this result, but is also flexible enough to analyse more complex sparsity connectivity patterns beyond Random Pruning.

This scaling directly influences network training dynamics. If $\lambda_k$ is the $k$-th eigenvalue of $\Theta_{\text{std}}$, then this $k$-th eigenvalue of the pruned network's NTK becomes $c^L \lambda_k$. Notably, while the absolute learning speed is reduced, the relative dynamics between modes remain unchanged. This offers a principled explanation for the empirical observation that sparse random networks converge more slowly than their dense counterparts [27]. We refer to Appendix B for more discussions.

## 6 Numerical experiments

We illustrate the relationship between spectral properties of Graphon NTK and training dynamics of finite sparse networks using three pruning methods: Random, SNIP [43], and Synflow [62] at

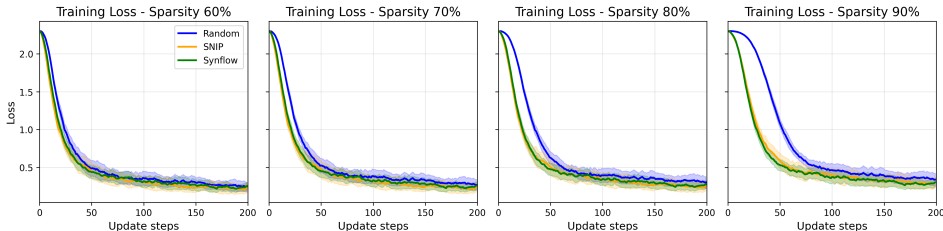

Figure 2: The training loss in the first 200 gradient update steps of training sparse networks produced by Random, SNIP, and Synflow with different sparsity levels. At the beginning of the training phase, subnetworks generated by SNIP and Synflow show a faster convergence speed than Random across sparsity levels.

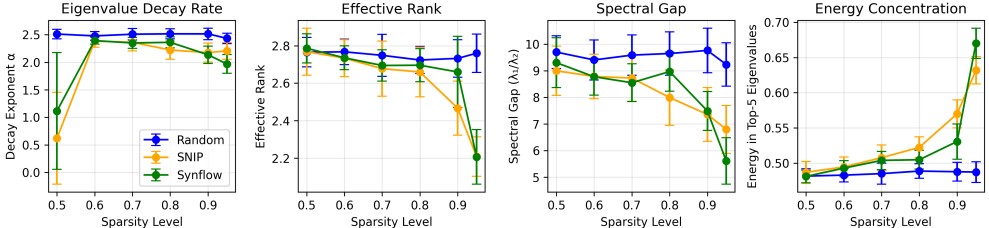

Figure 3: Spectral metrics of the Graphon NTK with different graphon functions and sparsity levels.

sparsity levels from 50% to 95%. Subnetworks are pruned from 4-layer networks with hidden size $n = 1024$, then trained on MNIST. We approximate the Graphon NTK by graphon functions found in Section 4. In particular, given the sparsity, we generate masks for 4-layer networks with hidden dimension $n = 1024$ based on graphon functions. Then we compute the Graphon NTK based on a batch of 1024 data samples from 10 classes in MNIST, and analyse four spectral metrics: eigenvalue decay rate ($\alpha$), effective rank trace($\Theta$)$/\lambda_1$, spectral gap ($\lambda_1/\lambda_2$), and the energy concentration in top-5 eigenvalues $\frac{\sum_{i=1}^{5} \lambda_i}{\sum_{j=1}^{n} \lambda_j}$. We refer to Appendix D for further details.

**Results and discussion.** Figure 3 reveals that Random Pruning maintains relatively consistent spectral properties across the sparsity levels, with stable decay rate, high effective rank, and broad spectral spread. This reflects uniform eigenvalue scaling which is consistent with the constant graphon analysis in Section 5.3, where Random Pruning acts as a global downscaling of the kernel. In contrast, SNIP and Synflow increasingly concentrate their Graphon NTK energy in top eigenvalues as the sparsity level grows, despite reduced effective rank and spectral gaps. This suggests a stronger focus on dominant eigen-directions, aligning with their faster training loss reduction at the beginning when compared with random subnetworks at the same sparsity in Figure 2.

The observed correlation between kernel spectral properties and training dynamics demonstrates an initial insight of our Graphon NTK framework for sparse neural network training. These findings offer both theoretical insight and practical guidance: Graphon NTK can serve as principled, training-free indicators for evaluating pruning quality, and preserving key spectral characteristics should be a design goal for future pruning algorithms, as also verified in [67].

# 7 Conclusion

In this paper, we introduce a novel theoretical framework for analysing sparse neural networks through the lens of graph limit theory and neural tangent kernel. Our Graphon Limit Hypothesis establishes a connection between pruning methods and their limiting graphons in the infinite-width regime, providing a mathematical framework for understanding the structural properties of sparse networks. We derive Graphon NTK which offers a meaningful tool for analysing how these structural properties affect training dynamics of sparse networks. This paves the way for the theoretical study of sparse models using tools from kernel and graph limit theories.

# 8   Limitations and future research directions

Our work introduces the Graphon NTK as a new theoretical tool for understanding neural network pruning. While it establishes a solid foundation, several limitations remain, and each suggests promising avenues for future research.

**Formalising the Graphon Limit Hypothesis**. A foundational aspect of our work is its reliance on the Graphon Limit Hypothesis, which is the proposition that pruning masks converge to a limit graphon. A key limitation is that we establish this hypothesis empirically, providing experimental evidence across MLP architectures and pruning methods, rather than through a formal mathematical proof. The theoretical framework of the Graphon NTK, which we use to study training dynamics and spectral properties, is consequently built upon this compelling, yet unproven, foundation. This immediately opens a critical avenue for future theoretical research to formally prove the existence and uniqueness of these graphon limits.

**From Static Analysis to Dynamic Pruning and Complex Architectures**. Our analysis currently focuses on the clean theoretical setting of pruning at initialisation using fixed masks on MLPs. This controlled environment was essential for developing the core theory but does not yet encompass more complex, real-world scenarios. This limitation presents a clear research trajectory which extends the framework to dynamic pruning and advanced architectures. For iterative or continuous sparsification, one can envision modelling the pruning process as an evolution of the graphon over time, potentially recasting the problem as a gradient flow on the continuous space of graphons. A parallel effort will be to generalise the Graphon NTK beyond MLPs to architectures like CNNs and Transformers.

**Bridging the Theory-Practice Gap with Principled Algorithm Design**. While our framework provides a powerful lens for analysing existing pruning methods, a key limitation is the inherent gap between its infinite-width theoretical predictions and the practical realities of finite-width networks. Furthermore, our analysis explains why certain structures are effective but does not yet yield a prescriptive algorithm for practitioners to create them. Future work can focus on creating graphon-guided pruning algorithms that directly optimise for a target graphon with desirable spectral properties (e.g., high energy concentration for faster convergence), thereby closing the loop from theory back to practice. Beyond just designing masks, these insights could inform graphon-informed sparse training methodologies, such as developing learning rate schedulers or weight initialisations based on the eigenfunctions of the Graphon NTK. Such methods would be designed to explicitly accelerate training and improve robustness, helping to bridge the theory-practice gap for finite networks.

# 9   Acknowledgements

RB acknowledges the Gauss Centre for Supercomputing e.V. for providing computing time on the GCS Supercomputer JUWELS at Jülich Supercomputing Centre (JSC). RB is also grateful for funding from the European Research Council (ERC) under the Horizon Europe Framework Programme (HORIZON) for proposal number 101116395 SPARSE-ML.

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

## Contents

# A  Details on graphon neural tangent kernel

In this section, we present a comprehensive derivation of the NTK for neural networks with graphon structure. This novel formulation extends the standard NTK theory to accommodate networks where connectivity patterns are modulated by graphon functions, providing insights into how network architecture influences learning dynamics. In particular, we first describe the network setting and the transition from discrete to continuous network in Appendix A.1. Then we provide the proof for the Proposition 1 in Appendix A.2 and for Theorem 1 in Appendix A.3, respectively.

## A.1  Network structure and setup

**Discrete neural network with graphon**  Consider a single output neural network with $L$ hidden layers where weights are modulated by a graphon function $\mathcal{W}^{(l)} : [0,1]^2 \to [0,1]$:

$$W_{ij}^{(l)} \sim \mathcal{N} \left( 0, \mathcal{W}^{(l)} \left( \frac{i}{n_l}, \frac{j}{n_{l-1}} \right) \right). \tag{5}$$

The graphon function governs the statistical structure and strength of synaptic connections between neurons. Specifically, it describes how the variance of random weights varies based on the positions of the connected neurons, thereby shaping the connectivity patterns and signal propagation across layers. Unlike in standard neural networks where weights are identically distributed, graphon-modulated weights have position-dependent variances while maintaining their independence.

Pre-activations and activations are computed as:

$$z_i^{(l)}(x) = \frac{1}{\sqrt{n_{l-1}}} \sum_{j=1}^{n_{l-1}} W_{ij}^{(l)} h_j^{(l-1)}(x), \tag{6}$$

$$h_i^{(l)}(x) = \sigma(z_i^{(l)}(x)). \tag{7}$$

The network output for simplicity is a single output:

$$f(x) = \frac{1}{\sqrt{n_L}} \sum_{j=1}^{n_L} W_{ij}^{(L+1)} h_j^{(L)}(x). \tag{8}$$

**Connection to network pruning**  The graphon framework can be directly connected to network pruning. In standard pruning, a binary mask $M^{(l)}$ is applied to the weights:

$$\widetilde{W}^{(l)} = W^{(l)} \odot M^{(l)}, \tag{9}$$

where $\odot$ denotes element-wise multiplication and $M_{ij}^{(l)} \in \{0,1\}$ indicates whether the connection is kept (1) or pruned (0). In the finite-width case, the variance of the pruned weights becomes:

$$\mathrm{Var}(\widetilde{W}_{ij}^{(l)}) = \sigma_w^2 M_{ij}^{(l)}, \tag{10}$$

which equals either $\sigma_w^2$ (kept) or 0 (pruned). As network width increases ($n_l, n_{l-1} \to \infty$), the pruning mask can be viewed as a step function over $[0,1]^2$, with $M_{ij}^{(l)}$ corresponding to the value at $\left( \frac{i}{n_l}, \frac{j}{n_{l-1}} \right)$. In the limit, this mask converges to the graphon $\mathcal{W}^{(l)}(u_l, u_{l-1})$, which represents the density or probability of connections in each region of the unit square. Different pruning methods yield different graphon structures. Thus, the graphon-modulated weights can be interpreted as modeling the effect of pruning directly within the network initialization, enabling analysis of pruned network behavior in the infinite-width limit.

**Continuous limit formulation**  As layer widths approach infinity ($n_l \to \infty$), we transition to continuous indices:

- $i/n_l \to u_l \in [0,1]$ (position in layer $l$),

- $j/n_{l-1} \to u_{l-1} \in [0, 1]$ (position in layer $l-1$).

The continuous network becomes:

$$z^{(l)}(u_l, x) = \int_0^1 W^{(l)}(u_l, u_{l-1}) h^{(l-1)}(u_{l-1}, x) du_{l-1}, \tag{11}$$

$$h^{(l)}(u_l, x) = \sigma(z^{(l)}(u_l, x)), \tag{12}$$

$$f(x) = \int_0^1 W^{(L+1)}(u_{L+1}, u_L) h^{(L)}(u_L, x) du_L, \tag{13}$$

where $W^{(l)}(u_l, u_{l-1}) \sim \mathcal{N}(0, \mathcal{W}^{(l)}(u_l, u_{l-1}))$.

**Statistical properties and convergence conditions**  In graphon-structured networks, weights become independent but not identically distributed (non-i.i.d.) random variables. This change requires careful consideration of the conditions under which the Law of Large Numbers (LLN) and Central Limit Theorem (CLT) apply.

*Law of Large Numbers in Graphon Networks*

For the LLN to hold in the non-i.i.d. setting of graphon networks, the following conditions are required:

- **Bounded Graphon Values**: $\mathcal{W}^{(l)}(u_l, u_{l-1}) \leq 1$ ensures that $\mathrm{Var}(W_{ij}^{(l)})$ is bounded, preventing any term from dominating the sum (satisfied by graphon definition).

- **Well-Defined Average Connectivity**: The integral $\int_0^1 \mathcal{W}^{(l)}(u_l, u_{l-1}) du_{l-1}$ must be well-defined for all positions $u_l$, ensuring the "average effect" of connections per neuron is stable.

Since the graphon values are bounded, the variances of weights and related quantities remain bounded, ensuring the LLN applies to empirical averages throughout the network.

*Central Limit Theorem and the Lindeberg-Feller Condition*

For pre-activations to converge to Gaussian processes in graphon networks, we assume the Lindeberg-Feller condition [6] is satisfied:

$$\lim_{n_{l-1} \to \infty} \frac{1}{\sigma_n^2} \sum_{j=1}^{n_{l-1}} \mathbb{E}\left[ X_j^2 \cdot \mathbf{1}_{\{|z_j| > \epsilon \sigma_n\}} \right] = 0, \tag{14}$$

for all $\epsilon > 0$, where $X_j = \frac{1}{\sqrt{n_{l-1}}} W_{ij}^{(l)} h_j^{(l-1)}(x)$, $\sigma_n^2 = \sum_{j=1}^{n_{l-1}} \mathrm{Var}(X_j)$, and the pre-activation $z_i = \sum_j^{n_{l-1}} X_j$. This condition ensures that no single term in the sum disproportionately influences the total variance. Then, the CLT applies despite the non-i.i.d. nature of the weights, allowing pre-activations to converge to Gaussian processes with position-dependent covariance structures.

In our setting, the Lindeberg condition is satisfied due to the following structural properties of graphon networks: (i) When network weights are initialized using a bounded graphon ($\mathcal{W}^{(l)}(u, v) \leq 1$), each connection's variance is controlled, preventing any single weight from becoming dominant; (ii) Activations from the previous layer $h_j^{(l-1)}(x)$ are bounded with sigmoid and tanh function, while activations like ReLU tend to remain within reasonable ranges when networks are properly initialized and inputs are normalized; (iii) The scaling factor of $\frac{1}{\sqrt{n_{l-1}}}$ in the pre-activation formula ensures that individual pre-activation variances decrease proportionally as the network widens, scaling as $O(\frac{1}{n_{l-1}})$. Together, these properties imply that as the layer width $n_{l-1} \to \infty$, the influence of any single term $X_j$ becomes negligible relative to the total variance. Hence, the Lindeberg condition is met, and the pre-activations converge in distribution to a Gaussian process with a position-dependent covariance structure induced by the graphon.

## A.2 Forward propagation: covariance structure (proof of proposition 1)

We first establish the statistical behaviour of signals as they propagate through the network. Our main result for the forward pass is captured in the Proposition 1. For the sake of convenience, we re-state the proposition here again:

**Proposition 2.** *For a neural network with layers structured by graphons $\mathcal{W}^{(l)} : [0,1]^2 \to [0,1]$, Lipschitz nonlinearity $\sigma$, and in the limit as $n_1, ..., n_L \to \infty$, the pre-activations $z^{(l)}(u_l, x)$ at every hidden layer converge to centred Gaussian processes with covariance $\tilde{\Sigma}^{(l)}$, where $\tilde{\Sigma}^{(l)}$ is defined recursively by:*

$$\tilde{\Sigma}^{(1)}(u_1, u_1', x, x') = \delta(u_1 - u_1') \frac{1}{d} \sum_{j}^{d} \mathcal{W}^{(1)}(u_1, \frac{j}{d})(x \cdot x')_j, \tag{15}$$

$$\tilde{\Sigma}^{(l)}(u_l, u_l', x, x') = \delta(u_l - u_l') \int_0^1 \mathcal{W}^{(l)}(u_l, u_{l-1}) \Sigma^{(l-1)}(u_{l-1}, u_{l-1}, x, x') du_{l-1}, \tag{16}$$

*where $(x \cdot x')_j$ represents the input correlation at position $j$, and the activation covariance $\Sigma^{(l)}$ is:*

$$\Sigma^{(l)}(u_l, u_l', x, x') = \delta(u_l - u_l') \mathbb{E}_{(z,z') \sim \mathcal{N}(0, \Lambda^{(l)}(u_l))}[\sigma(z)\sigma(z')], \tag{17}$$

*where $\delta(u_l - u_l')$ is Dirac delta function, $\Lambda^{(l)}(u_l)$ is the position-dependent covariance matrix:*

$$\Lambda^{(l)}(u_l) = \begin{bmatrix} \tilde{\Sigma}^{(l)}(u_l, u_l, x, x) & \tilde{\Sigma}^{(l)}(u_l, u_l, x, x') \\ \tilde{\Sigma}^{(l)}(u_l, u_l, x', x) & \tilde{\Sigma}^{(l)}(u_l, u_l, x', x') \end{bmatrix}.$$

*Proof:*

We prove the proposition by induction on the layer index $l$. The key insight is to analyze how the graphon structure affects the statistical properties of pre-activations as signals propagate through the network. The proof works as follows:

**Base case: first layer $l = 1$**   For the first layer with graphon modulation:

$$W_{ij}^{(1)} \sim \mathcal{N}\left(0, \mathcal{W}^{(1)}\left(\frac{i}{n_1}, \frac{j}{d}\right)\right), \tag{18}$$

where $d$ is the input dimension. The pre-activation covariance becomes:

$$\mathbb{E}[z_i^{(1)}(x) z_{i'}^{(1)}(x')] = \mathbb{E}\left[\left(\frac{1}{\sqrt{d}} \sum_{j=1}^{d} W_{ij}^{(1)} x_j\right)\left(\frac{1}{\sqrt{d}} \sum_{k=1}^{d} W_{i'k}^{(1)} x_k'\right)\right] \tag{19}$$

$$= \frac{1}{d} \sum_{j=1}^{d} \sum_{k=1}^{d} \mathbb{E}[W_{ij}^{(1)} W_{i'k}^{(1)}] x_j x_k'. \tag{20}$$

Since weights are independently sampled:

- $\mathbb{E}[W_{ij}^{(1)} W_{i'k}^{(1)}] = 0$ when $(i, j) \neq (i', k)$ ,
- $\mathbb{E}[(W_{ij}^{(1)})^2] = \mathcal{W}^{(1)}\left(\frac{i}{n_1}, \frac{j}{d}\right)$ when $i = i'$ and $j = k$ .

Therefore:

$$\mathbb{E}[z_i^{(1)}(x) z_{i'}^{(1)}(x')] = \delta_{ii'} \frac{1}{d}\left[\sum_{j=1}^{d} \mathcal{W}^{(1)}\left(\frac{i}{n_1}, \frac{j}{d}\right) x_j x_j'\right]. \tag{21}$$

As $n_1 \to \infty$ in the continuous limit with $i/n_1 \to u_1$:

$$\tilde{\Sigma}^{(1)}(u_1, u_1', x, x') = \delta(u_1 - u_1')\frac{1}{d}\sum_j^d \mathcal{W}^{(1)}(u_1, \frac{j}{d})(x \cdot x')_j, \tag{22}$$

where $\cdot$ is the dot-product, $(x \cdot x')_j$ represents the input correlation at position $j$.

The activation covariance at position $u_1$ is:

$$\Sigma^{(1)}(u_1, u_1', x, x') = \delta(u_1 - u_1')\mathbb{E}[\sigma(z^{(1)}(u_1, x))\sigma(z^{(1)}(u_1, x'))]. \tag{23}$$

Since $(z^{(1)}(u_1, x), z^{(1)}(u_1, x'))$ follows a joint Gaussian distribution according to the CLT, this can be computed as:

$$\Sigma^{(1)}(u_1, u_1', x, x') = \delta(u_1 - u_1')\mathbb{E}_{(z,z')\sim\mathcal{N}(0,\Lambda^{(1)}(u_1))}[\sigma(z)\sigma(z')] \tag{24}$$

Where $\Lambda^{(1)}(u_1)$ is the position-dependent covariance matrix:

$$\Lambda^{(1)}(u_1) = \begin{bmatrix} \tilde{\Sigma}^{(1)}(u_1, u_1, x, x) & \tilde{\Sigma}^{(1)}(u_1, u_1, x, x') \\ \tilde{\Sigma}^{(1)}(u_1, u_1, x', x) & \tilde{\Sigma}^{(1)}(u_1, u_1, x', x') \end{bmatrix}. \tag{25}$$

Different from fully connected network, where the pre-activation covariance of the first layer becomes covariance of the input layer $\Sigma^{(0)}(x, x') = x.x'$, in graphon networks, the pre-activation covariance depends on the structure of the graphon.

**Inductive step: subsequent layer $l > 1$**   For layer $l$ with graphon structure:

$$\mathbb{E}[z_i^{(l)}(x)z_{i'}^{(l)}(x')] = \mathbb{E}\left[\frac{1}{n_{l-1}}\left(\sum_{j=1}^{n_{l-1}} W_{ij}^{(l)}h_j^{(l-1)}(x)\right)\left(\sum_{k=1}^{n_{l-1}} W_{i'k}^{(l)}h_k^{(l-1)}(x')\right)\right] \tag{26}$$

$$= \frac{1}{n_{l-1}}\sum_{j=1}^{n_{l-1}}\sum_{k=1}^{n_{l-1}}\mathbb{E}[W_{ij}^{(l)}W_{i'k}^{(l)}]\mathbb{E}[h_j^{(l-1)}(x)h_k^{(l-1)}(x')]. \tag{27}$$

Since weights are independently sampled and $\mathbb{E}[(W_{ij}^{(l)})^2] = \mathcal{W}^{(l)}\left(\frac{i}{n_l}, \frac{j}{n_{l-1}}\right)$:

$$\mathbb{E}[z_i^{(l)}(x)z_{i'}^{(l)}(x')] = \delta_{ii'}\left[\frac{1}{n_{l-1}}\sum_{j=1}^{n_{l-1}}\mathcal{W}^{(l)}\left(\frac{i}{n_l}, \frac{j}{n_{l-1}}\right)\mathbb{E}[h_j^{(l-1)}(x)h_j^{(l-1)}(x')]\right]. \tag{28}$$

In the continuous limit as $j/n_{l-1} \to u_{l-1}$, and $1/n_{l-1}$ is absorbed into the integral:

$$\tilde{\Sigma}^{(l)}(u_l, u_l', x, x') = \delta(u_l - u_l')\int_0^1 \mathcal{W}^{(l)}(u_l, u_{l-1})\Sigma^{(l-1)}(u_{l-1}, u_{l-1}, x, x')du_{l-1}. \tag{29}$$

The activation covariance is:

$$\Sigma^{(l)}(u_l, u_l', x, x') = \delta(u_l - u_l')\mathbb{E}_{(z,z')\sim\mathcal{N}(0,\Lambda^{(l)}(u_l))}[\sigma(z)\sigma(z')], \tag{30}$$

where $\Lambda^{(l)}(u_l)$ is the covariance matrix for pre-activations at position $u_l$:

$$\Lambda^{(l)}(u_l) = \begin{bmatrix} \tilde{\Sigma}^{(l)}(u_l, u_l, x, x) & \tilde{\Sigma}^{(l)}(u_l, u_l, x, x') \\ \tilde{\Sigma}^{(l)}(u_l, u_l, x', x) & \tilde{\Sigma}^{(l)}(u_l, u_l, x', x') \end{bmatrix}. \tag{31}$$

This completes the proof of Proposition 1 by induction. A critical insight is how the graphon structure $\mathcal{W}^{(l)}(u_l, u_{l-1})$ directly modulates signal propagation, creating non-uniform information flow across different network regions.

### A.3 Graphon neural tangent kernel convergence (proof of theorem 1)

Our main theoretical result characterises the NTK for graphon-structured networks. For convenience, we re-state the Theorem 1 here again:

**Theorem 2** (Graphon NTK). *For a neural network with layers structured by graphons $\mathcal{W}^{(l)}$ : $[0,1]^2 \to [0,1]$, Lipschitz nonlinearity $\sigma$, in the limit as $n_1, ..., n_L \to \infty$, the Graphon Neural Tangent Kernel (Graphon NTK) $\Theta(x, x')$ converges to a deterministic kernel:*

$$\Theta(x, x') = \sum_{l=1}^{L} \int_0^1 \left( \dot{\Sigma}^{(l)}(u_l, u_l, x, x') \int_{[0,1]^{L-l+1}} \prod_{m=l+1}^{L+1} \mathcal{W}^{(m)}(u_m, u_{m-1}) \dot{\Sigma}^{(m)}(u_m, u_m, x, x') \, d\mathbf{u}_{l+1} \right) \tag{32}$$

$$\left( \int_0^1 \Sigma^{(l-1)}(u_{l-1}, u_{l-1}, x, x') du_{l-1} \right) du_l$$

*where $\dot{\Sigma}^{(l)}(u_l, u_l, x, x') = \mathbb{E}[\sigma'(z^{(l)}(u_l, x))\sigma'(z^{(l)}(u_l, x'))]$ represents the expected correlation between activation derivatives, and $d\mathbf{u}_{l+1} = du_{L+1}du_L \ldots du_{l+1}$.*

*Proof:*

We prove the Graphon NTK by decomposing the derivation into three key steps: (1) characterizing gradient flow through graphon-structured layers, (2) analyzing gradient correlations under infinite-width statistical independence, and (3) integrating these correlations to derive a closed-form NTK expression. This structured approach yields an interpretable kernel that reflects how the graphon structure shapes learning dynamics across the network.

#### A.3.1 Backward propagation: gradient flow

**Gradient recursion**  For layer $L$, applying the chain rule:

$$\frac{\partial f(x)}{\partial z^{(L)}(u_L, x)} = \frac{\partial f(x)}{\partial h^{(L)}(u_L, x)} \frac{\partial h^{(L)}(u_L, x)}{\partial z^{(L)}(u_L, x)} = W^{(L+1)}(u_{L+1}, u_L)\sigma'(z^{(L)}(u_L, x)). \tag{33}$$

For earlier layers ($l < L$), the gradient with respect to pre-activations is:

$$\frac{\partial f(x)}{\partial z^{(l)}(u_l, x)} = \sigma'(z^{(l)}(u_l, x)) \int_0^1 \frac{\partial f(x)}{\partial z^{(l+1)}(u_{l+1}, x)} W^{(l+1)}(u_{l+1}, u_l) du_{l+1}. \tag{34}$$

**Parameter gradients**  The gradients with respect to weights are:

$$\frac{\partial f(x)}{\partial W^{(l)}(u_l, u_{l-1})} = \frac{\partial f(x)}{\partial z^{(l)}(u_l, x)} h^{(l-1)}(u_{l-1}, x). \tag{35}$$

#### A.3.2 Graphon neural tangent kernel derivation

The Neural Tangent Kernel is defined as the inner product of gradients with respect to all parameters:

$$\Theta(x, x') = \sum_{l=1}^{L} \int_0^1 \int_0^1 \mathbb{E} \left[ \frac{\partial f(x)}{\partial W^{(l)}(u_l, u_{l-1})} \frac{\partial f(x')}{\partial W^{(l)}(u_l, u_{l-1})} \right] du_l du_{l-1}. \tag{36}$$

**Layer-wise graphon NTK contribution**  The contribution to the Graphon NTK from layer $l$ is:

$$\Theta_l(x, x') = \int_0^1 \int_0^1 \mathbb{E} \left[ \frac{\partial f(x)}{\partial W^{(l)}(u_l, u_{l-1})} \frac{\partial f(x')}{\partial W^{(l)}(u_l, u_{l-1})} \right] du_l du_{l-1} \tag{37}$$

$$= \int_0^1 \int_0^1 \mathbb{E} \left[ \frac{\partial f(x)}{\partial z^{(l)}(u_l, x)} h^{(l-1)}(u_{l-1}, x) \frac{\partial f(x')}{\partial z^{(l)}(u_l, x')} h^{(l-1)}(u_{l-1}, x') \right] du_l du_{l-1}. \tag{38}$$

At this point, we apply a key insight: In the infinite-width limit, by the LLN, the gradients $\frac{\partial f(x)}{\partial z^{(l)}(u_l, x)}$ and the activations $h^{(l-1)}(u_{l-1}, x)$ become statistically independent for distinct positions $u_l$ and $u_{l-1}$. This allows us to factorize the expectation:

$$\Theta_l(x, x') = \int_0^1 \mathbb{E}\left[\frac{\partial f(x)}{\partial z^{(l)}(u_l, x)}\frac{\partial f(x')}{\partial z^{(l)}(u_l, x')}\right]\left(\int_0^1 \mathbb{E}[h^{(l-1)}(u_{l-1}, x)h^{(l-1)}(u_{l-1}, x')]du_{l-1}\right)du_l \tag{39}$$

$$= \int_0^1 \mathbb{E}\left[\frac{\partial f(x)}{\partial z^{(l)}(u_l, x)}\frac{\partial f(x')}{\partial z^{(l)}(u_l, x')}\right]\left(\int_0^1 \Sigma^{(l-1)}(u_{l-1}, u_{l-1}, x, x')du_{l-1}\right)du_l. \tag{40}$$

**Gradient correlation analysis** To compute the NTK, we need to analyze the correlation between gradients at different inputs:

$$\mathbb{E}\left[\frac{\partial f(x)}{\partial z^{(l)}(u_l, x)}\frac{\partial f(x')}{\partial z^{(l)}(u_l, x')}\right]. \tag{41}$$

From our backward propagation analysis in Equation 34, we have:

$$\mathbb{E}\left[\frac{\partial f(x)}{\partial z^{(l)}(u_l, x)}\frac{\partial f(x')}{\partial z^{(l)}(u_l, x')}\right] = \mathbb{E}\left[\sigma'(z^{(l)}(u_l, x))\sigma'(z^{(l)}(u_l, x'))\right.$$

$$\left.\int_0^1\int_0^1 \frac{\partial f(x)}{\partial z^{(l+1)}(u_{l+1}, x)}\frac{\partial f(x')}{\partial z^{(l+1)}(u'_{l+1}, x')}W^{(l+1)}(u_{l+1}, u_l)W^{(l+1)}(u'_{l+1}, u_l)du_{l+1}du'_{l+1}\right]. \tag{42}$$

In the infinite-width limit, $(z^{(l)}(u_l, x), z^{(l)}(u_l, x'))$ follows a bivariate Gaussian distribution, allowing us to define:

$$\dot\Sigma^{(l)}(u_l, u_l, x, x') = \mathbb{E}[\sigma'(z^{(l)}(u_l, x))\sigma'(z^{(l)}(u_l, x'))]. \tag{43}$$

The weights $W^{(l+1)}(u_{l+1}, u_l)$ and $W^{(l+1)}(u'_{l+1}, u_l)$ are independent for $u_{l+1} \neq u'_{l+1}$, with:

$$\mathbb{E}[W^{(l+1)}(u_{l+1}, u_l)W^{(l+1)}(u'_{l+1}, u_l)] = \mathcal{W}^{(l+1)}(u_{l+1}, u_l)\delta(u_{l+1} - u'_{l+1}). \tag{44}$$

Substituting this into our gradient correlation:

$$\mathbb{E}\left[\frac{\partial f(x)}{\partial z^{(l)}(u_l, x)}\frac{\partial f(x')}{\partial z^{(l)}(u_l, x')}\right] = \dot\Sigma^{(l)}(u_l, u_l, x, x') \tag{45}$$

$$\int_0^1 \mathcal{W}^{(l+1)}(u_{l+1}, u_l)\mathbb{E}\left[\frac{\partial f(x)}{\partial z^{(l+1)}(u_{l+1}, x)}\frac{\partial f(x')}{\partial z^{(l+1)}(u_{l+1}, x')}\right]du_{l+1}.$$

**Closed-form expression for Graphon NTK** To derive the closed-form expression, we define:

$$G^{(l)}(u_l, x, x') = \mathbb{E}\left[\frac{\partial f(x)}{\partial z^{(l)}(u_l, x)}\frac{\partial f(x')}{\partial z^{(l)}(u_l, x')}\right]. \tag{46}$$

From our derivation, $G^{(l)}$ follows the recursion:

$$G^{(l)}(u_l, x, x') = \dot\Sigma^{(l)}(u_l, u_l, x, x')\int_0^1 \mathcal{W}^{(l+1)}(u_{l+1}, u_l)G^{(l+1)}(u_{l+1}, x, x')du_{l+1}. \tag{47}$$

With the base case $G^{(L+1)}(u_{L+1}, x, x') = 1$.

The general form for any layer $l$ becomes:

$$G^{(l)}(u_l, x, x') = \dot{\Sigma}^{(l)}(u_l, u_l, x, x') \int_{[0,1]^{L-l+1}} \prod_{m=l+1}^{L+1} \mathcal{W}^{(m)}(u_m, u_{m-1}) \dot{\Sigma}^{(m)}(u_m, u_m, x, x') \, d\mathbf{u}_{l+1},$$

(48)

where $d\mathbf{u}_{l+1} = du_{L+1} du_L \dots du_{l+1}$ and $\dot{\Sigma}^{(L+1)}(u_{L+1}, u_{L+1}, x, x') = 1$.

Substituting this into our formula for $\Theta_l$:

$$\Theta_l(x, x') = \int_0^1 G^{(l)}(u_l, x, x') \left( \int_0^1 \Sigma^{(l-1)}(u_{l-1}, u_{l-1}, x, x') du_{l-1} \right) du_l \tag{49}$$

$$= \int_0^1 \left( \dot{\Sigma}^{(l)}(u_l, u_l, x, x') \int_{[0,1]^{L-l+1}} \prod_{m=l+1}^{L+1} \mathcal{W}^{(m)}(u_m, u_{m-1}) \dot{\Sigma}^{(m)}(u_m, u_m, x, x') \, d\mathbf{u}_{l+1} \right) \tag{50}$$

$$\left( \int_0^1 \Sigma^{(l-1)}(u_{l-1}, u_{l-1}, x, x') du_{l-1} \right) du_l. \tag{51}$$

The full Graphon NTK is the sum over all layers:

$$\Theta(x, x') = \sum_{l=1}^{L} \Theta_l(x, x') \tag{52}$$

$$= \sum_{l=1}^{L} \int_0^1 \left( \dot{\Sigma}^{(l)}(u_l, u_l, x, x') \int_{[0,1]^{L-l+1}} \prod_{m=l+1}^{L+1} \mathcal{W}^{(m)}(u_m, u_{m-1}) \dot{\Sigma}^{(m)}(u_m, u_m, x, x') \, d\mathbf{u}_{l+1} \right) \tag{53}$$

$$\left( \int_0^1 \Sigma^{(l-1)}(u_{l-1}, u_{l-1}, x, x') du_{l-1} \right) du_l \tag{54}$$

This expression reveals how the graphon structure at each layer shapes the Neural Tangent Kernel through multiple integrals involving the graphon functions. The resulting kernel is position-dependent and reflects the specific connectivity patterns encoded by the graphons.

### A.3.3 Discussion and implication

The Graphon NTK provides a powerful analytical framework for understanding how structured weight patterns influence neural network learning dynamics. Several key insights emerge:

1. **Non-uniform signal propagation**: The graphon structure creates position-dependent information flow, with regions of higher graphon values propagating signals more strongly.

2. **Relationship to pruning**: The graphon formulation provides a continuous limit perspective on network pruning, where the graphon $\mathcal{W}^{(l)}(u_l, u_{l-1})$ can be interpreted as the density or probability of connections.

3. **Position-dependent learning dynamics**: Different regions of the network effectively learn at different rates based on their connectivity patterns.

These theoretical results establish the foundation for analysing learning behaviours in neural networks with connectivity patterns, providing insights that may guide the development of more efficient architectural designs.

## B    Details on graphon neural tangent kernel of Random pruning

With Random pruning, pruning masks converge to constant graphons (the Erdős–Rényi random graph) as the width tends to infinity. When the underlying graphons are constants, we observe a uniform scaling effect on training dynamics.

For a constant graphon $W(u, v) = c$, the resulting NTK scales uniformly as:

$$\Theta(x, x') = c^L \, \Theta_{std}(x, x'), \tag{55}$$

where $\Theta_{std}^{(L)}$ denotes the standard NTK of a fully-connected network.

*Proof:*

For the first hidden layer ($l = 1$), pre-activation covariance becomes:

$$\tilde{\Sigma}^{(1)}(u, u', x, x') = \delta(u - u') \left( c(x \cdot x') \right). \tag{56}$$

For simplicity, assuming $\sigma_b^2 = 0$ (no biases):

$$\tilde{\Sigma}^{(1)}(u, u', x, x') = c \, \tilde{\Sigma}_{std}^{(1)}(u, u', x, x'). \tag{57}$$

This leads to activation covariance at first layer:

$$\Sigma^{(1)}(u, u', x, x') = c \, \Sigma_{std}^{(1)}(u, u', x, x'). \tag{58}$$

By induction, for any layer l:

$$\tilde{\Sigma}^{(l)}(u, u', x, x') = \delta(u - u') \left( c \int_0^1 \Sigma^{(l-1)}(v, v, x, x') dv \right) \tag{59}$$

$$= \delta(u - u') \left( c \int_0^1 c^{l-1} \, \Sigma_{std}^{(l-1)}(v, v, x, x') dv \right)$$

$$= \delta(u - u') \left( c^l \int_0^1 \Sigma_{std}^{(l-1)}(v, v, x, x') dv \right)$$

$$= c^l \, \tilde{\Sigma}_{std}^{(l)}(u, u', x, x').$$

And the activation covariance at layer l is:

$$\Sigma^{(l)}(u, u', x, x') = c^l \, \Sigma_{std}^{(l)}(u, u', x, x'). \tag{60}$$

We also assume that the activation function is ReLU, then the relationship between activation derivative covariance:

$$\dot{\Sigma}^{(l)}(x, x') = \dot{\Sigma}_{std}^{(l)}(x, x'). \tag{61}$$

For the gradient correlation function:

$$G^{(l)}(u, x, x') = \dot{\Sigma}^{(l)}(u, u, x, x') \int_0^1 \mathcal{W}^{(l+1)}(v, u) G^{(l+1)}(v, x, x') dv. \tag{62}$$

For a homogeneous graphon, each backward step contributes a factor of $c$. Going from output layer $L + 1$ back to layer $l$:

$$G^{(L+1)}(u, x, x') = G_{\text{std}}^{(L+1)}(u, x, x') = 1$$

$$G^{(L)}(u, x, x') = \dot{\Sigma}^{(L)}(u, u, x, x') \int_0^1 \mathcal{W}^{(L+1)}(v, u) G^{(L+1)}(v, x, x') dv$$

$$G^{(L)}(u, x, x') = c\,\dot{\Sigma}^{(L)}(u, u, x, x')$$

$$G^{(L)}(u, x, x') = c\,G_{\text{std}}^{(L)}(u, x, x')$$

$$G^{(L-1)}(u, x, x') = \dot{\Sigma}^{(L-1)}(u, u, x, x')\,c \int_0^1 G^{(L)}(v, x, x') dv$$

$$G^{(L-1)}(u, x, x') = c^2 G_{\text{std}}^{(L-1)}(u, x, x')$$

By induction, we have:

$$G^{(l)}(u, x, x') = c^{L+1-l}\,G_{\text{std}}^{(l)}(u, x, x'). \tag{63}$$

The layer-wise NTK contribution is:

$$\Theta_l(x, x') = \int_0^1 G^{(l)}(u, x, x') \left( \int_0^1 \Sigma^{(l-1)}(v, v, x, x') dv \right) du. \tag{64}$$

Substituting our findings:

$$\Theta_l(x, x') = \int_0^1 c^{L+1-l}\,\dot{\Sigma}^{(l)}(u, x, x') \left( \int_0^1 c^{l-1} \Sigma_{\text{std}}^{(l-1)}(v, v', x, x') dv \right) du. \tag{65}$$

When the activation integral dominates the constant 1 (which is typical in deep networks):

$$\Theta_l(x, x') = c^{L+1-l}\,c^{l-1}\,\Theta_{l,\text{std}}(x, x') = c^L\,\Theta_{l,\text{std}}(x, x'). \tag{66}$$

Summing over all layers:

$$\Theta(x, x') = \sum_{l=1}^{L} \Theta_l(x, x') = c^L \sum_{l=1}^{L} \Theta_{l,\text{std}}(x, x') = c^L\,\Theta_{\text{std}}(x, x'). \tag{67}$$

This uniform scaling directly impacts convergence rates during training. If $\lambda_k$ are the eigenvalues of $\Theta_{\text{std}}$, then the eigenvalues of $\Theta$ become $c^L \lambda_k$. Consequently, residual modes during training decay as:

$$a_k(t) = a_k(0) e^{-c^L \lambda_k t}. \tag{68}$$

This scaling directly influences network training dynamics. If $\lambda_k$ are the eigenvalues of $\Theta_{\text{std}}$, then the eigenvalues of the pruned network's NTK become $c^L \lambda_k$. Notably, while the absolute learning speed is reduced, the relative dynamics between modes remain unchanged. This offers a principled explanation for the empirical observation that sparse random networks converge more slowly than their dense counterparts [27].

More broadly, our framework extends beyond this homogeneous case, allowing for arbitrary position-dependent graphons that induce heterogeneous learning dynamics. This highlights the generality and flexibility of Graphon NTK in modeling structured sparsity and its effect on neural dynamics.

# C Experiments on graph limit of pruning at initialisation methods

We empirically validate the graphon hypothesis by examining whether pruning methods converge to distinct, characteristic graphons as network width increases. We analyze four pruning-at-initialization methods: Random pruning, SNIP [43], GraSP [65], and Synflow [62] across varying network widths $n \in \{100, 500, 1000, 2000\}$, layers $4, 5$, and sparsity levels $\{70\%, 80\%, 90\%\}$. We conduct 100 independent trials per configuration and exclude masks from input and output layers. We follow Synflow [62] source code to produce masks [2]. All the experiments are run on a single Nvidia A10 GPU (24GB).

To visualize the emergent graphons, we employ the SAS method [12] that:

1. Sorts nodes based on degree centrality (out-degree for layer $l$, in-degree for layer $l + 1$). For 2D histogram, nodes in layer $l$ and $l + 1$ are in the x-axis and y-axis, respectively.

2. Partitions the sorted bipartite graph into a grid of intervals. Each axis is split into $K = 64$ intervals.

3. Computes the average edge density within each interval.

This degree-based sorting serves as an approximate measure-preserving transformation, revealing underlying structural patterns while maintaining invariance to node permutations.

We visualise graph limits of subnetworks' masks produced by PaI methods at different sparsity levels in 4- and 5-layer networks in Figures 5, 6, 7, 8, and 9. In particular, Random pruning converges to a constant graphon (Erdős-Rényi random graph), with uniform connection probability across all node positions. SNIP and GraSP exhibit structured, non-uniform graphons with density gradients, preferentially connecting high-centrality nodes. Synflow converges to a block-like graphon with sharp transitions, strongly prioritising connections among high-centrality neurons. In Figure 9, a large part of neurons is eliminated in hidden layers creating a subnetwork with high paths. This observation is also indicated in [56, 54], in which after each iteration, Synflow prunes weights connected to low-degree nodes.

To quantify convergence, we show the Euclidean distance between density matrices at width $N$ and reference matrices at $N = 2000$ as in Figure 4. Since all histograms are aligned via degree-based sorting, we used the Euclidean distance between the density matrices as a proxy for the cut distance

$$d(W_N, W_{2000}) = \sqrt{\sum_{i,j} (H_N(i,j) - H_{2000}(i,j))^2}.$$

where $H$ is the histogram.

All methods demonstrate monotonic convergence, with distances decreasing as width increases, confirming that the limiting graphon structure is an intrinsic characteristic of each pruning algorithm.

These results provide compelling evidence that each pruning method induces a unique (up to relabelling), stable connectivity pattern in the large-width limit, validating our graphon hypothesis and establishing a foundation for analysing pruning methods through graph limit theory.

---

[2]https://github.com/ganguli-lab/Synaptic-Flow

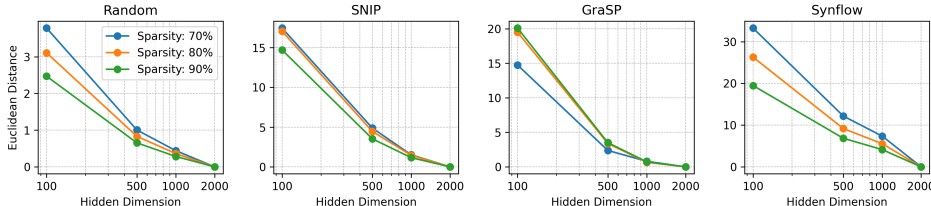

(a) Histogram convergence via Euclidean distance in 4-layer networks setting.

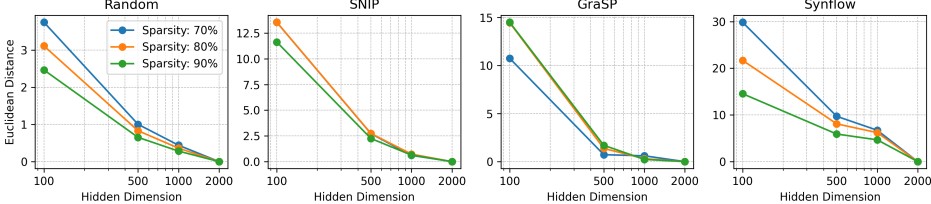

(b) Histogram convergence via Euclidean distance in 5-layer networks setting.

Figure 4: Histogram convergence via Euclidean distance.

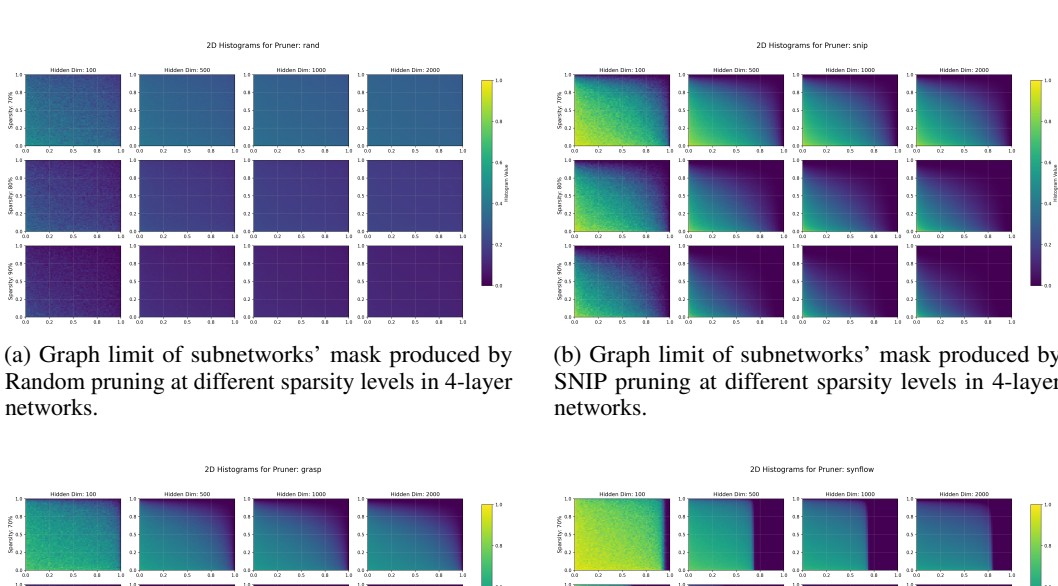

(a) Graph limit of subnetworks' mask produced by Random pruning at different sparsity levels in 4-layer networks.

(b) Graph limit of subnetworks' mask produced by SNIP pruning at different sparsity levels in 4-layer networks.

(c) Graph limit of subnetworks' mask produced by GraSP pruning at different sparsity levels in 4-layer networks.

(d) Graph limit of subnetworks' mask produced by Synflow pruning at different sparsity levels in 4-layer networks.

Figure 5: Graph limit of subnetworks' mask produced by PaI methods at different sparsity levels in 4-layer networks.

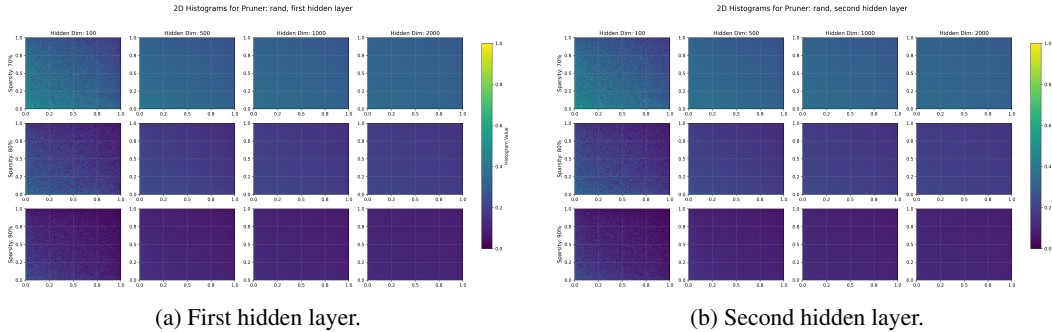

(a) First hidden layer.                    (b) Second hidden layer.

Figure 6: Graph limit of subnetworks' mask produced by Random pruning at different sparsity levels in 5-layer networks.

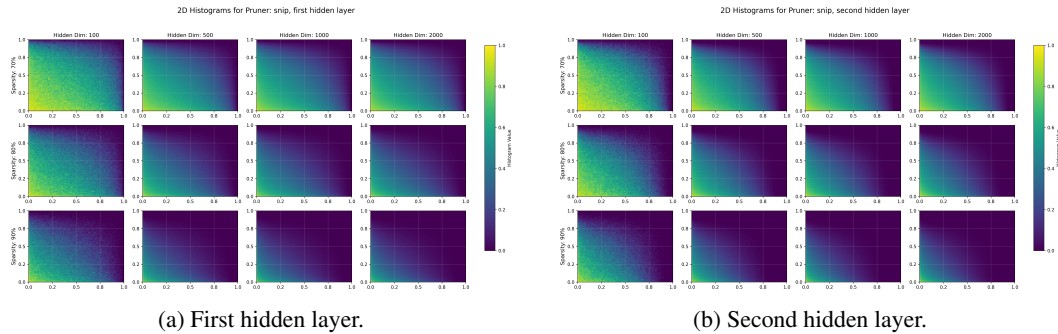

(a) First hidden layer.                    (b) Second hidden layer.

Figure 7: Graph limit of subnetworks' mask produced by SNIP pruning at different sparsity levels in 5-layer networks.

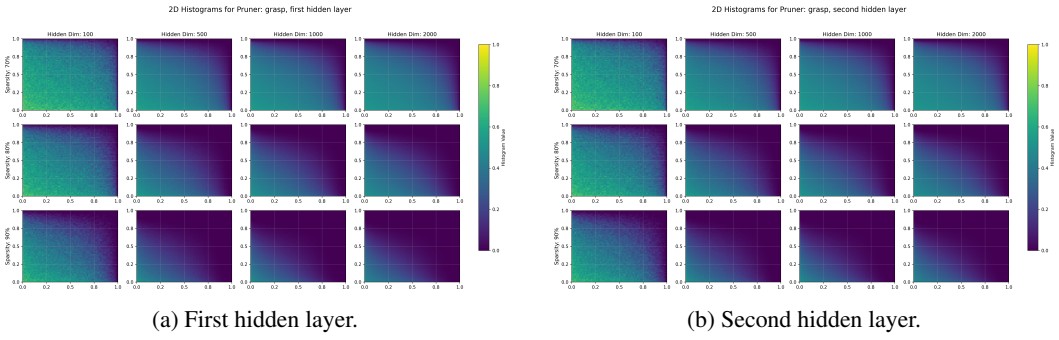

(a) First hidden layer.                    (b) Second hidden layer.

Figure 8: Graph limit of subnetworks' mask produced by GraSP pruning at different sparsity levels in 5-layer networks.

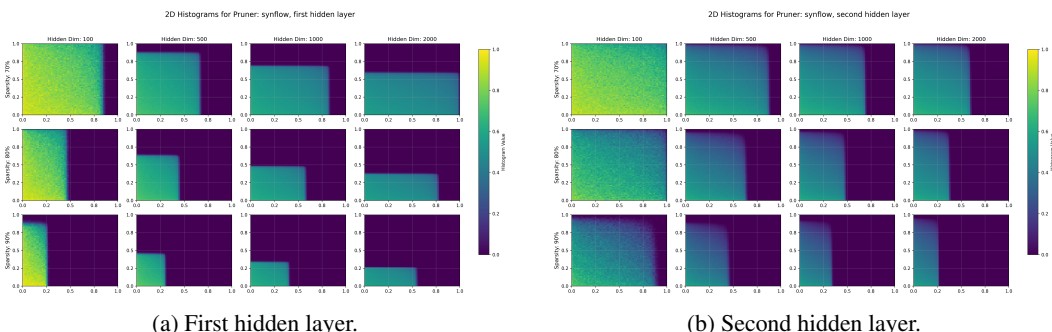

(a) First hidden layer.                    (b) Second hidden layer.

Figure 9: Graph limit of subnetworks' mask produced by Synflow pruning at different sparsity levels in 5-layer networks.

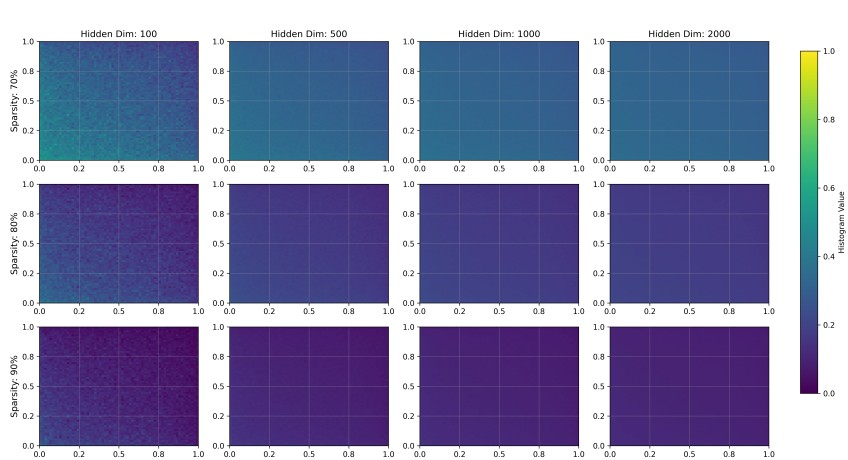

Figure 10: Graph limit of subnetworks' mask produced by Magnitude pruning at different sparsity levels in 4-layer networks.

**Magnitude Pruning at Initialisation.** Through the experiment, we find that Magnitude produces masks converge to constant graphons as shown in Figure 10. From a purely structural perspective, Magnitude pruning is identical to that of Random pruning in case all the weights are i.i.d. initialised. In particular, when all weights are i.i.d., the probability that any single weight $w_i j$ has a magnitude $|w_i j| > Threshold$ is the same for all $(i, j)$. Let this probability of preserving a connection be $c$. By definition, a graph where every edge exists with an independent and identical probability $c$ is an Erdős–Rényi random graph, $G(n, c)$. The well-known limit of a sequence of Erdős–Rényi graphs is a constant graphon $\mathcal{W}(u, v) = c$.

# D    Details on numerical experiments

**Experimental setup**    We evaluate the relationship between Graphon NTK spectral properties and sparse networks training dynamics using three pruning methods: Random Pruning, SNIP [43], and SynFlow [62] at sparsity levels from 50% to 95%. Subnetworks are pruned from 4 hidden layers network with width $n = 1024$, then trained on MNIST. Subnetworks are then trained on MNIST with Adam Optimizer with $0.001$ learning rate. All experiments are run 3 times. We illustrate the training loss in the first 200 update steps during training.

We approximate the Graphon NTK by graphon functions provided in Section 4. In particular, given the sparsity, we generate the mask for a 4 hidden layers network with width $n = 1024$ based on graphon functions. Then we compute the Graphon NTK based on a batch of 1024 data samples from 10 classes in MNIST. Since the eigenvalues of the NTK (on the training data) quantify how much the kernel emphasizes certain directions in function space, we analyze four spectral metrics based on eigenvalues:

- Eigenvalue decay rate ($\alpha$): The decay exponent $\alpha$ describes how fast these eigenvalues fall off, and it is approximated by fitting into the pow-law curve $\lambda_k \propto k^{-\alpha}$.

- Effective rank trace$(\Theta)/\lambda_1$: Quantifies the functional "dimensionality" of the kernel, showing how many independent directions meaningfully contribute to learning; lower values indicate stronger concentration on a few key patterns

- Spectral gap ($\lambda_1/\lambda_2$): The ratio between the first and second eigenvalues, revealing how dominant the primary learning direction is compared to others; larger gaps indicate the network will prioritize one function class extensively.

- Energy concentration $\frac{\sum_{i=1}^{k} \lambda_i}{\sum_{j=1}^{n} \lambda_j}$: Measures what fraction of the kernel's total power resides in the top eigenvalues; higher concentration means the kernel heavily emphasizes a few key patterns. We use $k = 5$ in our experiments.

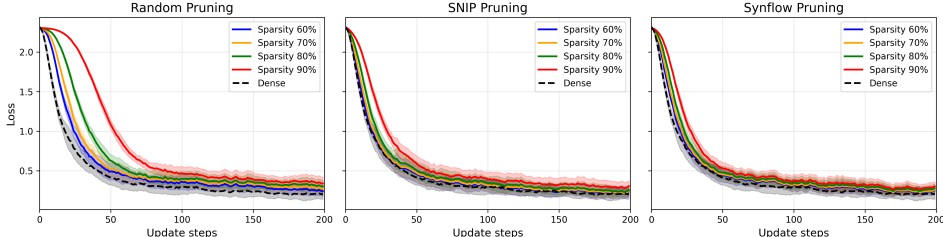

Figure 11: The training loss in the first 200 gradient update steps of training sparse networks produced by Random, SNIP, and Synflow with different sparsity levels compared with dense networks.

The training curves in Figure 11 reveal distinctive learning dynamics across pruning methods that can be elegantly explained through the Graphon NTK framework. For all methods, increased sparsity generally slows convergence, with dense networks typically converging fastest, but the magnitude of this effect varies significantly between approaches. Random pruning exhibits the strongest degradation with increased sparsity, showing a clear separation between different sparsity levels. Meanwhile, SNIP and Synflow maintain better convergence than random pruning at equivalent sparsity levels. These differences emerge from how each pruning method shapes the graphon structure ($\mathcal{W}^{(l)}$ functions) in the kernel formula.

