# OpenReview forum: "The Graphon Limit Hypothesis: Understanding Neural Network Pruning via Infinite Width Analysis"
_NeurIPS.cc/2025/Conference — NeurIPS 2025 spotlight_

### Official Review · Reviewer_xsUw · 2025-06-28

**Clarity:** 3
**Significance:** 3
**Originality:** 3
**Rating:** 5
**Confidence:** 2

**Summary:**

This paper introduces a theoretical framework for analyzing sparse neural networks obtained through various pruning methods. The authors link pruning masks to graphons in the infinite-width limit to examine connectivity patterns induced by different pruning techniques. Building on this, they propose Graphon NTK to study the training dynamics and spectral properties of sparse networks. Overall, this paper is well-structured and provides compelling experimental results.

**Questions:**

- The authors validate their method under only a single architecture and dataset. I am curious whether consistent patterns shown in the experiments appear when using different datasets or advanced architectures like transformers.
- I am also curious about the graph limit and spectral results of magnitude-based pruning approach.

**Ethical Concerns:**

["NO or VERY MINOR ethics concerns only"]

**Final Justification:**

After the rebuttal, I decide to maintain my rating in favor of acceptance. The authors provide rigorous and well-reasoned responses, along with clear explanations and additional experimental results that address most of my concerns.

Resolved issues include clarification of the practical relevance, relation to GNN pruning, application to dynamic pruning, and magnitude pruning analysis. However, the proposed work remains limited to MLPs, which constrains its broader applicability.

Nevertheless, I would like to assign more weight to the originality of the authors' contributions and in-depth rebuttal responses. I believe this paper makes self-contained contribution for future principled pruning analysis.

**Limitations:**

yes

**Paper Formatting Concerns:**

There are no formatting issues.

**Quality:**

4

**Strengths And Weaknesses:**

**Strengths**

- The connection between graphon limit theory and pruning mask is sound and novel. I believe this work would be a valuable contribution as analyzing pruning behaviors has been still underexplored.
- The experimental results are convincing, particularly the distinct differences in graph limit and spectral results between random pruning and other methods.

**Weaknesses**

- While the paper introduces an elegant theoretical tool to analyze pruning methods, it does not translate this framework into clear practical guidelines. For instance, it remains unclear how one can leverage graphon NTK analyses to select or design pruning strategies in real-world scenarios. The paper would benefit from connecting its theoretical insights to concrete, practical contributions.
- It would be interesting to add a discussion comparing the proposed graphon limit analysis with edge-pruning methods [1-3] in the GNN domain. Although the purpose is different, analyzing pruning masks as continuous adjacency matrices in latent space shares conceptual similarities with edge pruning, where one explicitly removes edges from the graph adjacency matrix in data space.
- If I understand correctly, the proposed framework does not appear to be applicable to dynamic pruning [4-5], which may limit its applicability.

**References**

---

[1] Chen, Tianlong, et al. "A unified lottery ticket hypothesis for graph neural networks." ICML 2021

[2] Seo, Hyunjin, Jihun Yun, and Eunho Yang. "TEDDY: Trimming edges with degree-based discrimination strategy." ICLR 2024

[3] Zhang, Guibin, et al. "Graph lottery ticket automated.” ICLR 2024

[4] Evci, Utku, et al. "Rigging the lottery: Making all tickets winners.”, ICML 2020

[5] Kusupati, Aditya, et al. "Soft threshold weight reparameterization for learnable sparsity." ICML 2020

---

> ### Author Rebuttal · Authors · 2025-07-30
>
> We thank the reviewer for your constructive feedback and recognition of our work. Below are our responses to the potential concerns and questions:
>
> **Q1**: “While the paper introduces an elegant theoretical tool to analyze pruning methods, it does not translate this framework into clear practical guidelines. For instance, it remains unclear how one can leverage graphon NTK analyses to select or design pruning strategies in real-world scenarios. The paper would benefit from connecting its theoretical insights to concrete, practical contributions.”
>
> **A1**: Regarding the choice of pruning strategies, our work lays the necessary groundwork for a more principled approach. By understanding how a method's graphon shapes the NTK's spectrum (as shown in Figure 3), one can move towards *designing pruning strategies by optimising for a target graphon* that yields desirable spectral properties (e.g., high energy concentration for faster convergence). This moves the field beyond pure heuristics. Recently, NTK spectral properties have already been leveraged to design pruning methods that preserve the eigenspectrum of “Dense” NTK [6]. It is expected that our Graphon NTK limit will also enable optimisation of sparse networks’ configurations and lead to concrete pruning methods.
>
> Furthermore, our analysis opens the door to designing more efficient training schemes for sparse networks. For example, one could design learning rate schedulers or weight initialisations based on the eigenfunctions of the Graphon NTK to further accelerate training for a given sparse architecture. However, identifying how specific weights contribute to particular eigenfunctions is inherently non-trivial due to the non-linearity of networks. While detailed algorithms for these are left for future work, our framework establishes its theoretical feasibility.
>
> **Q2**: “It would be interesting to add a discussion comparing the proposed graphon limit analysis with edge-pruning methods [1-3] in the GNN domain. Although the purpose is different, analyzing pruning masks as continuous adjacency matrices in latent space shares conceptual similarities with edge pruning, where one explicitly removes edges from the graph adjacency matrix in data space.”
>
> **A2**: We thank the reviewer for this insightful suggestion. The conceptual parallel between our work and edge-pruning in GNNs is indeed interesting, and we will add a discussion to the related work section in the final version. The critical difference lies in the object of pruning. In GNN pruning [1-3], one prunes the edges of the input data graph to improve efficiency or performance on that graph. In our work, we analyse the pruning of the model's own weight connections, which form an implicit connectivity graph within the neural network architecture itself. While both can be viewed through the lens of graph theory, the former relates to data preprocessing/regularisation, while our work relates to finding efficient model architectures. This distinction is crucial, and exploring the transfer of ideas between these two domains is a promising direction.
>
> **Q3**: “If I understand correctly, the proposed framework does not appear to be applicable to dynamic pruning [4-5], which may limit its applicability.”
>
> **A3**: Our current analysis focuses on pruning at initialisation (PaI) because it provides the cleanest setting for developing the theory, fixed masks enable rigorous infinite-width analysis. Extending this framework to dynamic pruning [4,5] is also possible to carry out graphon NTK analysis, but it requires significant additional ideas and technical details. For example, we would need to handle weight-mask co-evolution, significantly complicating the analysis. We believe understanding the "base case" of fixed masks is essential before tackling time-varying masks.
>
> For dynamic sparsification, it has been observed that some dynamic sparse training methods evolve from random to scale-free connectivity [7]. From our perspective, this can be viewed as the graphon evolving from a constant to a structured, power-law pattern. One could even recast dynamic sparsification as a *gradient flow problem on the continuous space of graphons.* These extensions require significant theoretical advancements, which we believe are well worth pursuing.
>
> **Q4**: “The authors validate their method under only a single architecture and dataset. I am curious whether consistent patterns shown in the experiments appear when using different datasets or advanced architectures like transformers.”
>
> **A4**: Our choice of MLP architectures provides a controlled setting to clearly establish our central theoretical claims. Extending this framework to more complex architectures like CNNs or Transformers is a primary goal for future work, but it presents significant and non-trivial challenges. To draw an analogy, extending the original dense NTK [8] from MLPs to architectures with convolution or attention required significant new technical developments [9,10] . Similarly, deriving a Graphon NTK for these architectures would involve handling concepts like convolution (for CNNs) or data-dependent connectivity (for Transformers), each constituting a substantial research project. However, we believe the fundamental principle that pruning masks converge to graphon limits is a general one.
>
> **Q5**: “I am also curious about the graph limit and spectral results of magnitude-based pruning approach.”
>
> **A5**: We thank the reviewer for this insightful question. Through the experiment, we find that Magnitude produced masks converge to constant graphons. From a purely structural perspective, Magnitude Pruning is identical to that of Random Pruning in case all the weights are i.i.d. initialised. In particular, when all weights are i.i.d., the probability that any single weight $w_{ij}$ has a magnitude $|w_{ij}| > Threshold$ is the same for all (i, j). Let this probability of preserving a connection be $s$. By definition, a graph where every edge exists with an independent and identical probability $s$ is an Erdős–Rényi random graph, $G(n, s)$. The well-known limit of a sequence of Erdős–Rényi graphs is a constant graphon $\mathcal{W}(u, v) = s$. We will incorporate these results in the next version of our paper.
>
> However, Magnitude PaI is not just a masking operation, it is a *selection procedure that biases the initial weight distribution.* The surviving weights are from a truncated distribution with a strictly larger variance than the original. Hence, while Magnitude PaI’s graphons are identical to Random Pruning, a larger variance in initial weights leads to a larger variance in pre-activations, which in turn produces "stronger" kernels with larger eigenvalues. This helps subnetworks produced by Magnitude learn in low-rank spaces faster, leading to faster initial training compared to Random Pruning.
>
> To demonstrate this, in Table 1, 2, we provide new experimental results at 80% sparsity (we will provide full results in our next version since we can not provide additional files/urls for NeurIPS rebuttal this year). This trend is consistent across different sparsity levels.
>
> Table 1: Top 10 eigenvalues (values in log scale)
> | Method | 1 | 2 | 3 | 4 | 5 | 6 | 7 | 8 | 9 | 10 |
> | --- | --- | --- | --- | --- | --- | --- | --- | --- | --- | --- |
> | Random | 19.09 | 16.71 | 16.48 | 16.30 | 16.17 | 15.99 | 15.87 | 15.71 | 15.60 | 15.44 |
> | Magnitude | 23.16 | 20.79 | 20.59 | 20.43 | 20.31 | 20.14 | 20.03 | 19.79 | 19.71 | 19.59 |
> | SNIP | 20.23 | 18.03 | 17.90 | 17.73 | 17.51 | 17.31 | 17.13 | 17.02 | 16.89 | 16.76 |
>
> Table 2: Training loss
> | Method | 0 | 10 | 20 | 30 | 40 | 50 | 60 | 70 | 80 | 90 | 100 | 110 | 120 | 130 | 140 | 150 | 160 | 170 | 180 | 190 |
> | --- | --- | --- | --- | --- | --- | --- | --- | --- | --- | --- | --- | --- | --- | --- | --- | --- | --- | --- | --- | --- |
> | Random | 2.30 | 2.21 | 1.80 | 1.22 | 0.85 | 0.63 | 0.52 | 0.47 | 0.42 | 0.40 | 0.40 | 0.41 | 0.37 | 0.38 | 0.35 | 0.36 | 0.32 | 0.31 | 0.33 | 0.32 |
> | Magnitude | 2.30 | 2.16 | 1.67 | 1.10 | 0.76 | 0.58 | 0.49 | 0.45 | 0.40 | 0.38 | 0.38 | 0.39 | 0.35 | 0.36 | 0.34 | 0.34 | 0.31 | 0.30 | 0.31 | 0.30 |
> | SNIP | 2.31 | 1.85 | 1.12 | 0.74 | 0.59 | 0.51 | 0.43 | 0.41 | 0.40 | 0.37 | 0.34 | 0.32 | 0.32 | 0.30 | 0.28 | 0.27 | 0.28 | 0.27 | 0.25 | 0.24 |
>
>
> Although kernels from Magnitude-pruned networks have higher eigenvalues than SNIP, SNIP provides higher energy concentration in the top eigenvalues, as illustrated in Table 3, which helps SNIP learn faster than Magnitude
>
> Table 3: Energy Concentration in top eigenvalues
> | Method    | Top 5      | Top 10     |
> |-----------|:-------------------:|:-------------------:|
> | Random    | 0.4756 ± 0.0140     | 0.5404 ± 0.0139     |
> | Magnitude | 0.4673 ± 0.0097     | 0.5335 ± 0.0097     |
> | SNIP      | 0.4996 ± 0.0186     | 0.5758 ± 0.0171     |
>
>
> ### References
> [6] Wang et. al., 2023. "NTK-SAP: Improving neural network pruning by aligning training dynamics."
>
> [7] Mocanu et. al., 2018. "Scalable training of artificial neural networks with adaptive sparse connectivity inspired by network science.”
>
> [8] Jacot et. al., 2018. "Neural tangent kernel: Convergence and generalization in neural networks."
>
> [9] Arora et. al., 2019 "On exact computation with an infinitely wide neural net."
>
> [10] Greg Yang, 2020. "Tensor programs ii: Neural tangent kernel for any architecture."

---

> > ### Comment · Reviewer_xsUw · 2025-08-06
> > **Official Comment by Reviewer xsUw**
> >
> > I appreciate the efforts the authors made during the rebuttal period. I keep my original acceptance rating.

---

> > > ### Author Response · Authors · 2025-08-06
> > >
> > > We thank the reviewer once again for your positive evaluation and very constructive suggestions on our work.

---

### Official Review · Reviewer_xZi5 · 2025-07-02

**Clarity:** 4
**Significance:** 3
**Originality:** 4
**Rating:** 6
**Confidence:** 4

**Summary:**

This submission proposes to study the sequence of masks produced by a given
pruning method as the width of networks goes to infinity.
Authors put forward the hypothesis that each pruning method yields masks which
admit a limit (in the form of a graphon), somehow caracteristic of the pruning
method. The paper does not provide theoretical arguments hinting at the
existence of such limits, but proposes experimental evidence supporting this
claim for MLPs of various depths, across a wide range of pruning methods.

**Questions:**

What *are* the challenges in training sparse networks (cited as early as L2
in the abstract and throughout the paper) ? Are there really any ?
The claim on L4 that different pruning mechanisms yield slightly different results
is backed by evidence, but the true difficulty inherent to sparse network
does not seem to have such support, as best seen perhaps on the paper's own
experiments in Figure 2, where all training methods consistently reach nearly-identical
loss values.
The claim that particular subnetworks are both harder to train but as good
as the unpruned networks they are extracted from (popularized by Frankle 2019
under the "Lottery Ticket Hypothesis" name), has been contested ever since its
introduction, including the very same year, by Zhuang Liu et al (2019, "Rethinking
the value of network pruning") able to train any random sparse network, at
scales much larger than the initial Lottery Ticket experiments, and consistently so,
provided hyperparameters are correctly tuned.

Following up on the previous question, how are the hyperparameters selected in
the training experiments presented ? If one subscribes to the idea that very
sparse networks are not particularly harder to train, but on the contrary just
require a new hyperparameter sweep, the results of Figure 2 are insufficient
to reach conclusions on the loss of accuracy obtained per sparsity level.
Can you extend these experiments to at least include small variations around
the hyperparameters selected, and would those alter the conclusions regarding
spectral properties ?

If the submission is accepted, and if the camera-ready limit is increased
to ten pages, as was previously the case, it seems to me that Figures 8 & 9
are worth the space they will take in the main text, they offer more valuable
insights into qualitatively different shapes across layers.
However, regardless of their position, the size of these figures needs to be
increased, they are hardly readable in their current state and I highly doubt that
the legends would be readable in print.

**Ethical Concerns:**

["NO or VERY MINOR ethics concerns only"]

**Final Justification:**

This submission proposes new tools to analyze pruning methods in deep learning models, by observing that many pruning methods of the literature admit a specific kind of limit, defined through a graphon. The existence and qualitatively different behaviors of different pruning methods are supported by reproducible experiments, and results are consistent with the presented theory and intuition.

The theoretical justifications for the existence of the limit are not particularly strong, and the definition presented is restricted to multi-layer perceptrons, which could limit impact. However, MLPs do exhibit very similar behaviors to richer architectures under pruning, which make it a good model to develop new tools to understand precise effects of various pruning methods. The connection of the limit graphon-ntk with observed properties such as convergence speed at high sparsity is also backed by convincing evidence. This is largely sufficient to justify the use of this tool to understand the effects of structured pruning at large widths, and further research efforts to extend this construction to richer architectures.

The spectral analysis of the limit graphon-ntk and the visualization of the shape of the limit graphon have potential to impact theoretical understanding of neural network convergence, inform domain-specific choices of pruning methods by practitioners, and even help develop new pruning methods with specific properties such as ensuring fast convergence at high sparsities or preserving specific eigenspaces of tangent kernels after pruning.

**Limitations:**

The discussion of limitations is only present in the appendix, which is a
questionnable choice, and although the gap between finite widths are graphon
limits of infinite widths is addressed, the question of whether the experiments
with relatively small width are really representative of the near-infinite
width regime is left unaddressed.
The restriction of these methods to (essentially) MLPs is not stressed enough,
neither in the main text, nor in the appendix

The restrictions to bounded value and well-behaved connectivity to make graphon
limits definable are substantial, but clearly outlined in the text.

**Quality:**

4

**Strengths And Weaknesses:**

The hypothesis and the underlying intuition are clearly laid out,
with convincing a-priori arguments, and backed by a thorough litterature review.
The experiments cover a wide range of pruning methods, although the historical
methods such as magnitude-based and hessian-based pruning are absent,
and all results support not only the idea that there is indeed a limit graphon,
but additionally that this graphon is very informative, allowing for instance
to clearly distinguish the SynFlow method from earlier SNIP and GraSP algorithms.

The computation of the Graphon NTK is soundly carried out, although
the collision in the choice of name with a previous use of the same term
is unfortunate. The estimations of its spectrum yield valuable insights into
pruning methods, and both tools have a high potential to become standard
analysis tools in the design of new pruning algorithms.

If anything, the experiments are a little lacking in size. In Figure 1,
a width of 2000 is not particularly large for a network which additionally
does not have to be trained since all pruning methods are compared at initialization,
surely this is an experiment which can be pushed to 10^4 width, given a reasonable
computation budget.
In Figure 3, only 128 samples to estimate the NTK is also a much smaller
batch size than the limit of what should be reasonably practical even
without GPUs on a day-long experiment.
In Figure 2, the choice to include the whole training window
obscures the comparison of values obtained. Having one or many plots at 200 update
steps with performance vs sparsity for each method would have been more striking
and easier to read values from.
Appendix C does provide details on material and methods, but does not report
training times, limiting the evaluation of whether it is reasonable to ask
for experiments at larger scale.

I wholeheartedly support the publication of this work, this is clearly
among the top papers that I expect to read from the conference.

---

> ### Author Rebuttal · Authors · 2025-07-30
>
> We sincerely appreciate the reviewer for recognising our work. We are also truly grateful for your constructive feedback. Below are our answers to your concerns and questions:
>
> **Q1**: “In Figure 1, a width of 2000 is not particularly large, …, surely this is an experiment which can be pushed to 10^4 width?”; “Appendix C does provide details on material and methods, but does not report training times, limiting the evaluation of whether it is reasonable to ask for experiments at larger scale.”
>
> **A1**: We have pushed the size of the network to $10^4$, unfortunately, this year NeurIPS does not allow submitting additional files/links. Thus, we can not provide the visualisation of graphon convergence. To illustrate the convergence, in Table 1, we present the Euclidean distance between histograms, in which we consider a network with size $10^4$ as a reference at 80% sparsity. All methods show a decreasing trend as width increases.
>
> Regarding the complexity, the SAS [1] algorithm requires sorting neurons based on degree taking O(nlogn) and O(n^2) in terms of memory. In practice, given a sparsity and pruning method, it takes ~120 minutes for n=$10^4$ in our machine with Dual AMD EPYC 7443 CPU and Nvidia A10 GPU.
>
> Table 1: Euclidean distance between histograms taking n=10000 as reference at 80% sparsity.
> | Method   | 100   | 500   | 1000  | 2000 | 10000 |
> |----------|:-----:|:-----:|:-----:|:-----:|:-----:|
> | SNIP     | 17.16 | 4.52  | 1.52  | 0.21 | 0.00  |
> | GraSP    | 17.82 | 2.56  | 2.22  | 2.23 | 0.00  |
> | SynFlow  | 30.74 | 16.39 | 12.72 | 9.73 | 0.00  |
>
> **Q2**: “In Figure 3, only 128 samples to estimate the NTK is also a much smaller batch size than the limit of what should be reasonably practical”
>
> **A2**: We have scaled the samples to 1024 with a hidden size of 1024 and also 512 with a 2048 hidden size. We observe a similar trend in the spectral metrics, where SNIP and SynFlow have higher energy concentration at top eigenvalues as illustrated in Table 2,3 below. We will incorporate this new result into the next version of the manuscript.
>
> Table 2: Energy Concentration in top5 eigenvalues in setting 1024 samples and 1024 hidden size.
> | Method \ Sparsity (%) | 50 | 60 | 70 | 80 | 90 | 95 |
> |:----------------------|:---:|:---:|:---:|:---:|:---:|:---:|
> | Random | 0.4509 | 0.4584 | 0.4538 | 0.4756 | 0.4694 | 0.4644 |
> | SNIP | 0.4584 | 0.4789 | 0.4851 | 0.4996 | 0.5379 | 0.6027 |
> | SynFlow | 0.4641 | 0.4733 | 0.4853 | 0.4985 | 0.5121 | 0.6592 |
>
> Table 3: Energy Concentration in top5 eigenvalues in setting 512 samples and 2048 hidden size.
> | Method \ Sparsity (%) | 50 | 60 | 70 | 80 | 90 | 95 |
> |:----------------------|:---:|:---:|:---:|:---:|:---:|:---:|
> | Random | 0.4652 | 0.4641 | 0.4629 | 0.4641 | 0.4658 | 0.4709 |
> | SNIP | 0.4670 | 0.4671 | 0.4768 | 0.4957 | 0.5480 | 0.6064 |
> | SynFlow | 0.4597 | 0.4687 | 0.4942 | 0.4858 | 0.5154 | 0.6433 |
>
> **Q3**: “In Figure 2, the choice to include the whole training window obscures the comparison of values obtained. Having one or many plots at 200 update steps with performance vs sparsity for each method would have been more striking and easier to read values from”
>
> **A3**: We thank the reviewer for the suggestions. The reason we included the training window is to compare the convergence rate of networks produced by different pruning methods.
> We add a table to show the final training loss at step 200 versus the sparsity levels for each pruning method as below:
>
> Table 4: Loss at step 200 with different sparsity levels and PaI methods.
> | Method \ Sparsity (%) | 0 | 60 | 70 | 80 | 90 |
> | --- | :---: | :---: | :---: | :---: | :---: |
> | Random | _ | 0.24 ± 0.07 | 0.27 ± 0.07 | 0.30 ± 0.08 | 0.34 ± 0.08 |
> | SNIP | _ | 0.24 ± 0.06 | 0.25 ± 0.06 | 0.26 ± 0.06 | 0.31 ± 0.07 |
> | SynFlow | _ | 0.25 ± 0.05 | 0.26 ± 0.05 | 0.28 ± 0.05 | 0.30 ± 0.06 |
> | Dense | 0.21 ± 0.07 | _ | _ | _ | _ |
>
>
> Besides, we would like to refer the reviewer to Appendix D in the manuscript for the comparison of training curves of different sparsity levels for each pruning method. Those figures indicate that increasing sparsity generally slows the convergence.
>
> **Q4**: “What are the challenges in training sparse networks...? The claim that particular subnetworks are both harder to train but as good as the unpruned networks... has been contested... by Zhuang Liu et al (2019, 'Rethinking the value of network pruning') able to train any random sparse network... provided hyperparameters are correctly tuned.”
>
> **A4**: We thank the reviewer for bringing up this important work by Zhuang Liu et al [2], which has shown that even random sparse networks can be trained to high accuracy if hyperparameters are meticulously tuned for each specific sparsity level. Our claim about the 'challenge' of sparse training is not that it is impossible, but that structured sparsity, as induced by methods like SNIP and Synflow, leads to architectures that are inherently more trainable with standard, less-tuned hyperparameters.
>
> Our framework provides a theoretical explanation for this phenomenon:
>
> - Random Pruning: Yields a constant graphon, which uniformly scales down the NTK. This slows learning for all functional modes equally, which is also observed in [3].
> - Structured Pruning (e.g., SNIP, Synflow): Yields structured graphons that concentrate the NTK's energy into dominant eigenmodes. This means the network tends to learn key functions quickly, even with a standard set of hyperparameters.
>
> So, the 'challenge' is one of trainability and robustness to hyperparameters. Our Graphon NTK analysis helps explain why some sparse structures are inherently 'better' initialisations for training than others. We will revise our introduction to state this more precisely, referencing the work of [2] to provide proper context.
>
> **Q5**: “How are the hyperparameters selected in the training experiments presented?”, “Can you extend these experiments to at least include small variations around the hyperparameters selected, and would those alter the conclusions regarding spectral properties?”
>
> **A5**: For the experiments in Figure 2, we used a fixed, standard set of hyperparameters (e.g., Adam optimiser [4] with a learning rate (lr) of 0.001) across all pruning methods and sparsity levels. Our goal was not to find the peak possible performance for each sparse mask, which would indeed require a full hyperparameter sweep per setting. Instead, our goal was to compare the inherent trainability of the architectures produced by each method under a single, consistent training protocol.
>
> Follow your suggestion, we have run with different lr values and we report the training loss of the first 100 steps with two lr 0.01 and 0.005 in Table 5,6, respectively (due to space limit). The first (resp. last) two rows are 80% (resp. 90%) sparsity.
>
> Table 5: Training loss with 0.01 lr
> |  | 0    | 10   | 20   | 30   | 40   | 50   | 60   | 70   | 80   | 90   | 100  |
> | - |:----:|:----:|:----:|:----:|:----:|:----:|:----:|:----:|:----:|:----:|:----:|
> | Random  | 2.30 | 1.61 | 1.07 | 0.70 | 0.51 | 0.42 | 0.38 | 0.34 | 0.35 | 0.33 | 0.31 |
> | SNIP    | 2.31 | 1.61 | 0.98 | 0.67 | 0.51 | 0.42 | 0.36 | 0.33 | 0.32 | 0.33 | 0.30 |
> | Random  | 2.30 | 1.75 | 1.12 | 0.78 | 0.58 | 0.46 | 0.41 | 0.36 | 0.31 | 0.31 | 0.32 |
> | SNIP    | 2.31 | 1.53 | 0.93 | 0.63 | 0.49 | 0.40 | 0.34 | 0.32 | 0.34 | 0.30 | 0.28 |
>
> Table 6: Training loss with 0.005 lr
> |  | 0    | 10   | 20   | 30   | 40   | 50   | 60   | 70   | 80   | 90   | 100  |
> | - |:----:|:----:|:----:|:----:|:----:|:----:|:----:|:----:|:----:|:----:|:----:|
> | Random  | 2.30 | 1.70 | 1.09 | 0.76 | 0.56 | 0.45 | 0.39 | 0.36 | 0.31 | 0.30 | 0.31 |
> | SNIP    | 2.31 | 1.48 | 0.91 | 0.60 | 0.46 | 0.40 | 0.33 | 0.31 | 0.30 | 0.28 | 0.26 |
> | Random  | 2.30 | 2.01 | 1.25 | 0.83 | 0.61 | 0.51 | 0.45 | 0.41 | 0.36 | 0.34 | 0.35 |
> | SNIP    | 2.31 | 1.52 | 0.93 | 0.64 | 0.50 | 0.43 | 0.36 | 0.32 | 0.31 | 0.29 | 0.28 |
>
>
> Although increasing lr speeds up the training of subnetworks, results from Table 5,6 still align with Figure 2, which show clear differences in convergence speed between Random and SNIP, therefore directly support our argument: with hyperparameters held constant, the network's structural properties (as encoded by the graphon) become the important factor in its training dynamics.
>
> **Q6**: “It seems to me that Figures 8 & 9 are worth the space they will take in the main text, they offer more valuable insights into qualitatively different shapes across layers. … the size of these figures needs to be increased, they are hardly readable ….”
>
> **A6**: We are grateful for these suggestions. We will definitely improve the visualisation and incorporate it into the next version.
>
> **Q7**: Absence of magnitude-based pruning methods
>
> **A7**: We have run additional experiments with Magnitude Pruning at Initialisation (PaI). Through the experiment, we find that Magnitude produced masks converge to constant graphons. From a purely structural perspective, Magnitude Pruning is identical to that of Random Pruning in case all the weights are i.i.d. initialised. In particular, when all weights are i.i.d., the probability that any single weight $w_{ij}$ has a magnitude $|w_{ij}| > Threshold$ is the same for all (i, j). Let this probability of preserving a connection be $s$. By definition, a graph where every edge exists with an independent and identical probability $s$ is an Erdős–Rényi random graph, $G(n, s)$. The well-known limit of a sequence of Erdős–Rényi graphs is a constant graphon $\mathcal{W}(u, v) = s$. We will incorporate these results into our next version.
>
>
> ### References
>
> [1] Chan et. al., 2014 "A consistent histogram estimator for exchangeable graph models."
>
> [2] Zhuang Liu, et. al., 2019 "Rethinking the value of network pruning."
>
> [3] Yang et. al., 2023. "On the neural tangent kernel analysis of randomly pruned neural networks.”
>
> [4] Kingma et. al., 2014. "Adam: A method for stochastic optimization."

---

> > ### Author Response · Authors · 2025-08-08
> >
> > Dear Reviewer,
> >
> > As the discussion period is approaching its deadline. We would appreciate knowing if our responses have addressed your concerns. If you have any remaining questions or concerns, please don't hesitate to share them with us. We will be happy to respond! Thank you for your time and recognition of our work.
> >
> > Bests,

---

> > > ### Comment · Reviewer_xZi5 · 2025-08-08
> > >
> > > All my questions have been addressed, and I'm happy to see that the original claims are further supported by the new measurements at higher scale and across variations of learning rates. Regarding answer A3, if the purpose of Fig 2 is primarily to support the observation that training speed varies rather than a comparison of final values, then maybe a log-log scale may be better suited, to better distinguish the initial drops and delays, and either way a comment interpreting the result in the caption could clarify the author's intention.
> > >
> > > I cannot increase my rating past 6, but I stand by my initial evaluation.

---

> > > > ### Author Response · Authors · 2025-08-08
> > > >
> > > > We are glad that our response has satisfactorily addressed the reviewer's questions. We would also like to thank the reviewer for your constructive and valuable feedback. We will incorporate your suggestions into the next version of our manuscript.

---

### Official Review · Reviewer_Z3zQ · 2025-07-03

**Clarity:** 3
**Significance:** 3
**Originality:** 3
**Rating:** 5
**Confidence:** 3

**Summary:**

This paper introduces a novel theoretical framework for analyzing sparse neural networks. The authors argue that such pruned networks exhibit unique structural properties in the infinite-width regime, depending on the pruning method. To analyze these structures, they propose a Graphon Neural Tangent Kernel framework, which captures how structural differences affect training dynamics. Empirical results show that popular pruning methods like SNIP and SynFlow increasingly concentrate their NTK energy on top eigenvalues as sparsity grows, suggesting a focus on dominant eigendirections. This aligns with their faster initial training loss reduction compared to random pruning at the same sparsity level.

**Questions:**

- Could the authors elaborate on why they chose to focus on pruning at initialization? Is this primarily due to compatibility with the NTK assumptions (infinite-width, lazy training)? Would the proposed framework extend to post-training pruning or iterative sparsification?
- The paper notes that SNIP and GraSP yield structured, non-uniform graphons, while SynFlow leads to block-like graphons that emphasize connections between high-centrality neurons. Could the authors elaborate on the intuitive explanation behind these observations? Are these structures a direct result of the pruning criteria used, and what does that imply?
- The authors use Euclidean distance between density matrices as a proxy for cut distance. Could they provide more justification or intuition for why this is a reliable proxy in this context? Are there limitations or assumptions that make this approximation valid?

**Ethical Concerns:**

["NO or VERY MINOR ethics concerns only"]

**Final Justification:**

This is an interesting and novel theory paper. It provides a useful tool for studying the structure and training dynamics of sparse networks. The authors' response fully resolved my concerns. I recommend its acceptance.

**Limitations:**

yes

**Quality:**

3

**Strengths And Weaknesses:**

Strengths
- The paper provides a novel theoretical tool (Graphon NTK) that offers insight into the structure and training dynamics of sparse networks.
- It bridges the gap between structural properties of pruned networks and their training behavior, helping explain why some pruning at initialization methods outperform random pruning even at high sparsity levels.
- The alignment between theory and empirical findings (e.g., energy concentration in top eigenvalues) strengthens the validity of the proposed framework.

Weaknesses
- The analysis is built upon the infinite-width NTK regime, which is known to diverge from practical finite-width networks in many important ways. Besides, in practice, the pruning is conducted on finite-width networks instead of infinite-width ones.
- The focus on pruning at initialization limits the applicability of the theory to real-world pruning workflows, which often involve iterative or post-training pruning.

---

> ### Author Rebuttal · Authors · 2025-07-30
>
> We thank the reviewer for your constructive feedback and for recognising the novelty and quality of our work. Below are our point-to-point responses to concerns and questions:
>
> **Q1**: “The analysis is built upon the infinite-width NTK regime, which is known to diverge from practical finite-width networks in many important ways. Besides, in practice, the pruning is conducted on finite-width networks instead of infinite-width ones.”
>
> **A1**: It is true that the NTK regime diverges from practical finite-width settings. However, NTK analysis is one of the only few available approaches for theoretical analysis of neural networks. Its value has been demonstrated recently in practice, for instance $\mu$P methods [1, 2, 3] for hyperparameter transfer.
>
> Our framework follows this spirit. By introducing the Graphon NTK, we provide a new, theoretically sound approach for analysing the training dynamics of sparse neural networks, analogous to what the standard NTK provides for dense networks. It opens up several promising practical research directions. For example, one could develop a $\mu$P-like transfer method for sparse networks or leverage spectral properties of Graphon NTK to accelerate the training of sparse networks.
>
> **Q2**: “The focus on pruning at initialization limits the applicability of the theory to real-world pruning workflows, which often involve iterative or post-training pruning.”; “Could the authors elaborate on why they chose to focus on pruning at initialization? Is this primarily due to compatibility with the NTK assumptions (infinite-width, lazy training)? Would the proposed framework extend to post-training pruning or iterative sparsification?”
>
> **A2**: We focused on pruning at initialisation (PaI) because it provides the cleanest setting for developing the theory, fixed masks enable rigorous infinite-width analysis. Extending this framework to dynamic or post-training pruning is also possible to carry out Graphon NTK analysis, but it requires significant additional ideas and technical details. For example, we would need to handle weight-mask co-evolution, significantly complicating the analysis. We believe understanding the "base case" of fixed masks is essential before tackling time-varying masks.
>
> In particular, with iterative pruning, one could study the *evolution of the graphon W(u, v, t)* at each pruning step t. Our framework provides the tools to analyse the spectral properties of the network at each stage, allowing one to ask: Which evolutionary paths for the graphon preserve trainability? For Dynamic Sparsification, it has been observed that some dynamic sparse training methods evolve from random to scale-free networks [4]. From our perspective, this can be viewed as the graphon evolving from a constant to a structured, power-law pattern. One could even recast dynamic sparsification as a *gradient flow problem on the continuous space of graphons.* These extensions require significant theoretical advancements, which we believe are well worth pursuing
>
> **Q3**: “The paper notes that SNIP and GraSP yield structured, non-uniform graphons, while SynFlow leads to block-like graphons that emphasize connections between high-centrality neurons. Could the authors elaborate on the intuitive explanation behind these observations? Are these structures a direct result of the pruning criteria used, and what does that imply?”
>
> **A3**: The emergent graphon structures are a reflection of each pruning method’s underlying philosophy:
>
> - SNIP and GraSP are two pruning methods based on the gradient/Hessian information on a batch of sample data. Therefore, neurons which have high gradient magnitudes or preserve second-order information will be kept, creating non-uniform and “hub”-like graphon patterns.
> - Synflow is an iterative data-agnostic pruning method which maximises the sum of path strengths. As a result, at each iteration, Synflow removes edges connecting to low-degree neurons, gradually narrowing the network  [5,6]. Hence, networks produced by Synflow converge to block-like graphons.
>
> **Q4**: “The authors use Euclidean distance between density matrices as a proxy for cut distance. Could they provide more justification or intuition for why this is a reliable proxy in this context? Are there limitations or assumptions that make this approximation valid?”
>
> **A4**: The cut distance quantifies the similarity between two graph adjacency matrices or functions by considering all possible labellings of their nodes. It requires finding optimal alignment between graphons, which is computationally intractable (NP-hard).
>
> We follow a degree-based alignment algorithm, sorting-and-smoothing (SAS) [7], which includes two main steps:
>
> (1) Alignment, where we sort neurons in each layer by their degree. This provides a canonical alignment that is provably near-optimal for graphons with monotone degree sequences [7]. For pruning methods that inherently rank connections by 'importance', this degree-based sorting is particularly effective and leads to structures with near-monotone degree sequences.
>
> (2) Distance computation where the cut distance between graphs is directly related to the Euclidean distance between their corresponding density matrices after alignment. Density matrices are constructed by partitioning sorted graphs into a grid of intervals, then computing the average edge density within each interval.
>
> By doing this, we are no longer comparing arbitrary node labellings but are instead comparing the aligned structures of graphs. The Euclidean distance then measures the difference in edge density between structurally equivalent regions. While this is not formally the cut distance, it is a highly reliable proxy for measuring differences in large-scale connectivity patterns, which is precisely the goal of our analysis.
>
> ### References
> [1] Yang et. al., 2020. "Feature learning in infinite-width neural networks."
>
> [2] Yang et. al., 2022 "Tensor programs v: Tuning large neural networks via zero-shot hyperparameter transfer."
>
> [3] Blake et. al., 2025 "u-$\mu $ P: The Unit-Scaled Maximal Update Parametrization."
>
> [4] Mocanu et. al., 2018. "Scalable training of artificial neural networks with adaptive sparse connectivity inspired by network science.”
>
> [5] Pham et. al., 2023 "Towards data-agnostic pruning at initialization: what makes a good sparse mask?.”
>
> [6] Patil et. al., 2021 "Phew: Constructing sparse networks that learn fast and generalize well without training data."
>
> [7] Chan et. al., 2014 "A consistent histogram estimator for exchangeable graph models."

---

> > ### Comment · Reviewer_Z3zQ · 2025-08-04
> > **Reply to rebuttal**
> >
> > Thanks for the detailed explanation and clarification. My concerns have been addressed, and I would like to raise my score accordingly.

---

> > > ### Author Response · Authors · 2025-08-04
> > >
> > > We thank the reviewer for your positive feedback on our rebuttal and for increasing the score.

---

### Official Review · Reviewer_Q1h3 · 2025-07-21

**Clarity:** 2
**Significance:** 2
**Originality:** 3
**Rating:** 4
**Confidence:** 4

**Summary:**

This paper studies the infinite width limit of pruned neural networks by introducing Graphons to analyze the infinite width limit of pruning masks. The paper first shows that various pruning at initialization masks converge to corresponding graphons. Then, the paper introduces the continuous limit of pruned networks and shows that they converge to gaussian processes with covariances modulated by the graphon that defines the pruning. Finally the paper empirically measures how pruning affects the NTK.

**Questions:**

See weaknesses for more questions. My basic issue with the paper is in its current state it is not clear how computing the graphon NTK can inform the choice of pruning strategies.

Can the graphon NTK only be used to analyze pruning strategies at initialization? Or can it be adapted to pruning trained networks?

**Ethical Concerns:**

["NO or VERY MINOR ethics concerns only"]

**Final Justification:**

I appreciate the author's clarification about how the Graphon reshapes the covariance of the gaussian process at each layer. This was a gap in my understanding that was resolved by their rebuttal. While my other questions were not completely addressed, I believe this paper can be accepted as a foundation for answering the questions I raised.

I have updated my score accordingly.

**Limitations:**

Yes

**Quality:**

3

**Strengths And Weaknesses:**

Strengths: The idea of studying pruning through graphons in the infinite width limit is promising. The experiments in section 4 demonstrate that pruning masks indeed converge to graphons. The results in section 5 help demonstrate that NTK analysis can be adapted with graphons.

Weaknesses:
While the motivation is interesting, I do not think the theoretical/empirical benefits of graphon NTK analysis have been demonstrated in this paper.

1. While the graphons are measured empirically, the authors do not characterize them except for the trivial case of random pruning (which leads to a constant function). Deriving the graphons, and then connecting their spectral properties to the properties of the pruned NTK can lead to a comparative analysis of pruning strategies. How do we choose pruning strategies that "preserve key spectral characteristics", as claimed in the paper?

2. The appearance of $\delta(u_1 - u_1')$ in the graphon NTK covariances means that functions at $u_1$ and $u_1'$ are uncorrelated (which tracks with the idea that they are independent draws from the same GP). From this observation, It seems that the role of the graphon basically boils down to a constant multiplying the covariance of the original NTK. Can you illustrate with a clear example how the NTK and Graphon NTK will differ?

3. Section 6 performs empirical measurements of the spectral properties of the graphon NTK, but does not connect them to faster convergence rates, informing the choice of hyperparameters, etc.

---

> ### Author Rebuttal · Authors · 2025-07-30
>
> We thank the reviewer for your constructive feedback. Below are our point-to-point responses to your concerns and questions:
>
> **Q1**: “I do not think the theoretical/empirical benefits of graphon NTK analysis have been demonstrated in this paper.”, “While the graphons are measured empirically, the authors do not characterize them …“
>
> **A1**: The primary benefit of our framework is that it provides a **new theoretical foundation and a unified analytical tool** for studying sparse neural networks. To our knowledge, this is the first work to connect the structure of pruning masks to graph limit theory, postulating the Graphon Limit Hypothesis and providing empirical evidence for it (Section 4). We then derive the Graphon NTK (Section 5), which explicitly links the structure in the limit to training dynamics.
>
> While we do not provide closed-form expressions for the graphon limits of methods like SNIP or Synflow, this is a common challenge in deep learning theory where the objects of study are complex. Instead, our experiments in Figures 1 and Appendix C empirically reveal their distinct and stable structures, which is a crucial first step for any analysis.
>
> **Q2**: “How do we choose pruning strategies that "preserve key spectral characteristics", as claimed in the paper?”, “My basic issue with the paper is in its current state it is not clear how computing the graphon NTK can inform the choice of pruning strategies.”
>
> **A2**: Regarding the choice of pruning strategies, our work lays the necessary groundwork for a more principled approach. As an example of its utility, we have linked the spectral properties of the Graphon NTK to the trainability and convergence speed of training different masks. For instance, in the case of high energy concentration (and thus faster learning of dominant features), we expect faster convergence, as shown in Figure 3. Accordingly, we have explained why pruning is more difficult in the high sparsity regime, especially in the case of random pruning.
>
> Furthermore, by understanding how a method's graphon shapes the NTK's spectrum, one could also move towards *designing pruning strategies by optimising for a target graphon* that yields desirable spectral properties (e.g., high energy concentration for faster convergence). This moves the field beyond pure heuristics. Recently, NTK spectral properties have already been leveraged to design pruning methods that preserve the eigenspectrum of “Dense” NTK [1]. It is expected that our Graphon NTK limit will also enable optimisation of sparse networks’ configurations and lead to concrete pruning methods. While detailed algorithms for these are a substantial future work, our framework establishes its theoretical feasibility.
>
> **Q3**: “The appearance of $\delta(u_1, u’_1)$ in the graphon NTK covariances means that functions at $u_1$  and $u’_1$ are uncorrelated, …, It seems that the role of the graphon basically boils down to a constant multiplying the covariance of the original NTK. Can you illustrate with a clear example how the NTK and Graphon NTK will differ?”
>
> **A3**: The $\delta(u_1, u’_1)$ appears because in the continuous limit, pre-activations at different positions evolve independently due to the infinite-width limit. Specifically:
>
> **In standard NTK**: $\widetilde{\Sigma}^{(l)}(x,x') = \Sigma^{(l-1)}(x,x')$
>
> **In Graphon NTK** (Eq. 2 in paper):  $\widetilde{\Sigma}^{(l)}(u_l, u_l', x, x') = \delta(u_l - u_l') \int^1_0 \mathcal{W}^{(l)} (u_l, u_{l-1}) \Sigma^{(l-1)}(u_{l-1}, u_{l-1}, x, x') du_{l-1}$
>
> The key difference is that standard NTK has **homogeneous** covariance (same for all neurons), while Graphon NTK has **position-dependent** covariance induced by graphon. The $\delta$ function enforces that only neurons at the same position are correlated in the infinite-width limit. Interpretably, in dense networks, all neurons learn uniformly, while in graphon-structured networks, neurons at position u learn at a rate proportional to $\int \mathcal{W}(u,v)dv$
>
> **In summary, the graphon does not simply multiply the covariance, it reshapes it via integration.** The only case where it becomes a simple scalar multiplier is when the graphon itself is a constant (e.g., Random Pruning), which our framework correctly recovers the result from [2] as a special case. For any structured graphon, the resulting Graphon NTK is fundamentally different from the standard NTK.
>
> **Q4**: “Section 6 performs empirical measurements of the spectral properties of the graphon NTK, but does not connect them to faster convergence rates, informing the choice of hyperparameters, etc.”
>
> **A4**: The connection to convergence rate can be made more explicit as follows: (1) Figure 2 shows that networks pruned with SNIP and Synflow converge faster than those from Random Pruning; (2) Figure 3 indicates that the graphons from SNIP and Synflow produce Graphon NTKs that **concentrate more energy in their top eigenvalues** than Random Pruning; (3) and a kernel that concentrates more of its energy in its top eigenvalues will drive faster convergence, as these correspond to the dominant learnable functions [3].
>
> This analysis opens the door to designing more efficient training schemes for sparse networks. For example, one could design learning rate schedulers or weight initialisations based on the eigenfunctions of the Graphon NTK to further accelerate training for a given sparse architecture. While identifying the correspondence between specific weights and eigenfunctions is a non-trivial challenge, our work establishes the theoretical pathway.
>
> **Q5**: “Can the graphon NTK only be used to analyze pruning strategies at initialization? Or can it be adapted to pruning trained networks?”
>
> **A5**: We focused on pruning at initialisation (PaI) because it provides the cleanest setting for developing the theory. Fixed masks enable a rigorous infinite-width analysis. Extending our Graphon NTK analysis to dynamic or post-training pruning is also possible, but it requires significant additional ideas and technical details. For example, we would need to handle weight-mask co-evolution, significantly complicating the analysis. We believe understanding the "base case" of fixed masks is essential before tackling time-varying masks.
>
> Overall, we believe establishing rigorous theoretical foundations is valuable even without immediate algorithmic payoffs, as demonstrated by the impact of works like NTK theory and mean-field theory in deep learning. Our graphon framework provides a new lens for understanding sparse networks that we hope will inspire future algorithmic innovations.
>
> ### References
>
> [1] Wang et. al., 2023. "NTK-SAP: Improving neural network pruning by aligning training dynamics."
>
> [2] Yang et. al., 2023. "On the neural tangent kernel analysis of randomly pruned neural networks."
>
> [3] Kopitkov et. al., 2020. "Neural spectrum alignment: Empirical study."
>
> [4] Mocanu et. al., 2018. "Scalable training of artificial neural networks with adaptive sparse connectivity inspired by network science."

---

> > ### Author Response · Authors · 2025-08-08
> >
> > Dear Reviewer,
> >
> > As the discussion period is approaching its deadline. We would appreciate knowing if our responses have addressed your concerns. If you have any remaining questions or concerns, please don't hesitate to share them with us. We will be happy to respond! Thank you for your time and consideration.
> >
> > Bests,

---

### Decision · Program_Chairs · 2025-09-17

**Decision:**

Accept (spotlight)

**Comment:**

This paper proposes to study the sequence of masks produced by a given pruning method as the width of networks goes to infinity. The paper puts forward the hypothesis that some pruning methods yield masks which admit a limit in the form of a graphon, somehow characteristic of the pruning method. The paper does not provide theoretical arguments hinting at the existence of such limits, but proposes experimental evidence supporting this claim for MLPs of various depths, across a wide range of pruning methods. Under this hypothesis, the paper proposes a graphon NTK approach to study the training dynamics and spectral properties of networks under pruning.

Overall, the reviewers were consistently positive about this paper, praising the novel idea of analyzing pruning via graphon analysis, which is backed by elegant theory, and the supportive experimental results.

This is a high quality paper and a clear “accept.”